# Online Robust Reinforcement Learning Through Monte-Carlo Planning

**Tuan Dam** [* 1]  **Kishan Panaganti** [* 2]  **Brahim Driss** [* 3]  **Adam Wierman** [2]

## Abstract

Monte Carlo Tree Search (MCTS) is a powerful framework for solving complex decision-making problems, yet it often relies on the assumption that the simulator and the real-world dynamics are identical. Although this assumption helps achieve the success of MCTS in games like Chess, Go, and Shogi, the real-world scenarios incur ambiguity due to their modeling mismatches in low-fidelity simulators. In this work, we present a new robust variant of MCTS that mitigates dynamical model ambiguities. Our algorithm addresses transition dynamics and reward distribution ambiguities to bridge the gap between simulation-based planning and real-world deployment. We incorporate a robust power mean backup operator and carefully designed exploration bonuses to ensure finite-sample convergence at every node in the search tree. We show that our algorithm achieves a convergence rate of $\mathcal{O}(n^{-1/2})$ for the value estimation at the root node, comparable to that of standard MCTS. Finally, we provide empirical evidence that our method achieves robust performance in planning problems even under significant ambiguity in the underlying reward distribution and transition dynamics.

## 1. Introduction

Reinforcement learning (RL) provides a statistical machine learning framework to interact with the environments—such as autonomous vehicles, agile robots, and network systems—sequentially and learn to take control actions to achieve the desired objective. Monte Carlo Tree Search (MCTS) algorithm, in conjunction with deep learning methods, solve complex decision-making problems in high-dimensional environments. Its celebrated success stories include autonomous RL decision-making agents playing board games Chess, Go, Shogi (Silver et al., 2016; Schrittwieser et al., 2020), Poker (Brown and Sandholm, 2018; Keshavarzi and Navidi, 2025), and solving various real-world challenging tasks like robotics and autonomous systems (Hoel et al., 2019; Kartal et al., 2019; Dam et al., 2022). *MCTS offers a principled way to balance exploration and exploitation by using combinatorial search mechanisms derived from online simulated trajectories.* As a result, MCTS can effectively promote the exploration of promising regions of the environment with only partial modeling information of the environment.

However, most of these successes are limited to structured or simulated environments. As successful as RL algorithms are, an issue in applying them to real-world dynamical systems is the unavoidable discrepancy between the simulators and the actual real-world system dynamics. In traditional RL approaches (Kaelbling et al., 1996; Salvato et al., 2021), transition models are often learned from data collected by interacting with simulator models to avoid unsafe interactions with real-world systems, and reward models may be subject to stochasticity, hacked rewards, or unmodeled external factors. Such ambiguities arise from a variety of sources: limited training data, non-stationary environments, adversarial conditions, partial observability, or simply modeling simplifications. These factors can lead to a so-called *simulation-to-reality* gap, where the policy or value function that appears optimal in the simulated environment may perform poorly when deployed in the real world. A natural approach to addressing these challenges is to incorporate robustness against simulation-to-reality gaps directly into the planning algorithm.

*RL agents making decisions under the framework of Robust Markov Decision Processes (RMDPs) (Iyengar, 2005; Nilim and El Ghaoui, 2005) offer a principled mechanism to conceptualize robustness against transition model and reward model mismatches raised by simulation-to-reality gaps.* These *robust RL agents* explore policies that maximize expected returns under the worst-case model within a prescribed *ambiguity set*. The ambiguity set is typically constructed as a ball around the simulator dynamics or re-

*Equal contribution  [1]Hanoi University of Science and Technology, Hanoi, Vietnam  [2]Department of Computing and Mathematical Sciences, California Insitute of Technology Pasadena, CA, USA  [3]Univ. Lille, Inria, CNRS, Centrale Lille, UMR 9189-CRIStAL. Correspondence to: Tuan Dam <tuandq@soict.hust.edu.vn>.

*Proceedings of the $42^{nd}$ International Conference on Machine Learning*, Vancouver, Canada. PMLR 267, 2025. Copyright 2025 by the author(s).

ward model, with the design choice of the ball size covering the real-world ground truth model descriptors. Recent works demonstrate their potential to achieve robust decision-making performance when faced with perturbations in transition dynamics and reward function models. However, while value iteration and policy optimization methods have been introduced and analyzed for robust RL, MCTS-based planning algorithms have not been explored, as per the authors' knowledge. We discuss more detailed related works in Section 2.

In this work, we propose a novel robust MCTS algorithm equipped with non-asymptotic performance guarantees under model ambiguity set. Importantly, we incorporate both *reward* and *transition* ambiguity robustness, similar to recent works (Zhou et al., 2021; Wang et al., 2024b) in robust RL. In particular, our work resolves the following questions:

> *Can we use a search-based planning approach like MCTS to balance exploitation and exploration for the robust RL problem? What theoretical guarantee can we provide? Can we show robust performance against standard algorithms under the simulation-to-reality issue?*

Our approach embeds the *distributionally robust optimization* (Rahimian and Mehrotra, 2019) mathematical principle into the MCTS framework, ensuring that the value estimates and action selections are robust to transitions and rewards drawn from the ambiguity sets. More precisely, we conceptualize a robust backup operator and design exploration bonuses that accommodate ambiguity sets defined using total variation, Kullback-Leibler, chi-squared, or Wasserstein measures. This allows MCTS to simultaneously use a tree search mechanism to solve for *robust value estimates* by trading off exploitation and exploration while achieving robust policies that work uniformly well across different models in the ambiguity set.

One of the key contributions of this work is the establishment of finite-sample bounds on the convergence rates of our robust MCTS algorithm. Viewing each node in the MCTS tree as a non-stationary bandit problem sheds light on the nontrivial challenges of controlling the interaction between ambiguity sets and exploration bonuses. More specifically, coming up with exploration bonuses (thereby robust value approximations) is nontrivial based on the non-linear backup operator due to the formalization of robustness. We overcome these challenges by building on a sequence of technical lemmas and applying concentration inequalities to the robust backup operator, we show that our method attains a convergence rate of order $\mathcal{O}(n^{-1/2})$ for robust value estimation at the root node, where $n$ is the number of states visited while exploring the environment. This convergence rate also matches the best-known results

for standard, non-robust MCTS, thereby demonstrating that introducing robustness need not change the convergence speed in terms of the number of samples.

**Contributions.** In this work, to the best of our knowledge, we are the *first* to propose an MCTS-based algorithm for the robust RL problem. Our contributions are threefold:

- Robust MCTS Algorithm: We solve the *online robust RL problem*–accounting for model ambiguity in both transitions and rewards–using a planning algorithm enabled by MCTS. This fundamental first step paves the way for future applications in large-scale dynamical systems.
- Non-Asymptotic Guarantees: We provide rigorous finite-sample performance bounds, ensuring that the robust MCTS converges with a known rate, on par with standard MCTS. Our analysis leads to novel exploration bonuses that arise from careful analyses of robust backup operators and the tree search mechanism by recasting robust MCTS for different ambiguity sets as a collection of non-stationary multi-armed bandit problems.
- Robust Empirical Performance: We conduct experiments in two environments (Gambler's Problem and Frozen Lake) to evaluate our robust algorithm, demonstrating that it achieves superior robust performance to model mismatches than the standard MCTS algorithm baseline.

## 2. Related Works

**Robust RL.** Robust RL agents make decisions to alleviate environmental ambiguities under the RMDP framework introduced by Iyengar (2005); Nilim and El Ghaoui (2005) considers distributional robust optimization (Rahimian and Mehrotra, 2019) mathematical formularization. Many recent works extensively study the robust RL problem, addressing multiple aspects of the challenges of decision-making learning algorithms. Panaganti and Kalathil (2021); Zhou et al. (2021); Panaganti and Kalathil (2022); Shi and Chi (2024) propose model-based dynamic programming algorithms to solve the robust RL problem for finite state-action environments, and Dong et al. (2022); Panaganti et al. (2025) extend to the online and offline settings, respectively. These works focus on addressing the sample complexity—minimal samples needed from the simulator model (leading to the construction of an approximate model) *for every state-action pair* to obtain an approximate value estimation—issue. Panaganti and Kalathil (2021); Panaganti et al. (2022); Zhang et al. (2023) propose model-free value function approximation-based robust RL algorithms utilizing special structures in the Bellman backups arising due to specific forms of ambiguity sets. Different from these approaches, our algorithm is inspired by MCTS to solve the robust RL problem. MCTS scales well (Silver et al., 2016) for large problems by embedding strong search

mechanisms into model-based planning approaches in RL.

**MCTS for non-robust RL.** AlphaGo-like (Silver et al., 2016) agents are powered by tree search mechanisms such as MCTS in traditional dynamic programming planning for standard RL. Kocsis and Szepesvári (2006); Shah et al. (2020); Dam et al. (2024b) provide theoretical guarantees for such heuristic search-based deep RL algorithms. Recently, the adoption of MCTS (Świechowski et al., 2023) in other learning settings has seen scaling advantages. For instance, in non-standard RL settings, like supervised learning systems (Guez et al., 2018; Wang et al., 2024a), constrained dynamical systems (Parthasarathy et al., 2023; Kurečka et al., 2024) to promote safe decision-making choices, and partially observable and constrained dynamical systems (Lee et al., 2018; Dam et al., 2022; 2020). In bandits, like agents taking decisions in the space of contexts (Ontanón, 2013; Mao et al., 2020). In applications, like autonomous vehicles and robots, (Kartal et al., 2019; Yin et al., 2022) where the imitation of expert decisions plays a critical role. Alternative approaches include entropy regularization methods like MENTS (Xiao et al., 2019), RENTS and TENTS (Dam et al., 2021; 2024a), and Boltzmann-based approaches (Painter et al., 2023), though these rely on temperature parameters that may impede convergence to true optimal values. Inspired by such adoption of MCTS, we enable MCTS-based planning for the *first* time to the robust RL problem—equipped with theoretical guarantees—that accounts for mitigating dynamical model ambiguities.

**Search-based planning for online robust RL.** This line of research is closest to ours in terms of search-inspired algorithms. (Liu et al., 2022; Wang et al., 2023; Wang, 2024) introduces the Multi-Level Monte Carlo (MLMC) method (Heinrich, 2001; Giles, 2008) to approximate the robust Bellman backups. MLMC is another powerful statistical sampling method from the family of Monte Carlo estimators. However, they have the drawback of requiring random sampling procedures in each iteration of the robust RL planning stages for every state-action pair. By avoiding these pitfalls, MCTS adapts to the online sampling procedure by enabling search from a tree node—states and actions in dynamical systems—up to some constant depth in the tree. Other works introduce sampling-based Q-learning (Zhou et al., 2021; Liu et al., 2022; Wang et al., 2024b) and policy iteration (Panaganti and Kalathil, 2021; Kumar et al., 2023; Badrinath, 2023) inspired approaches. These are popular methods in standard online RL enabling trajectory-based updates—at current states, actions, and next states sampled with an updated policy—to approximate the Bellman backups. However, these require algorithmic and theoretical innovations–for e.g., function approximation architectures–for scaling up

to high-dimensional dynamical systems (Panaganti et al., 2022; Zhang et al., 2023; Panaganti et al., 2024; Liu and Xu, 2024). The incorporation of the strong sampling procedure by MCTS avoids this issue.

## 3. Preliminaries

A Markov Decision Process (MDP) specified by the tuple $(\mathcal{S}, \mathcal{A}, P, R)$, where $\mathcal{S} \subset \mathbb{R}^d$ is the (potentially large) state space, $\mathcal{A}$ is a discrete action space, $P : \mathcal{S} \times \mathcal{A} \to \Delta(\mathcal{S})$ is the transition model mapping each state–action pair to a probability distribution over next states, and $R : \mathcal{S} \times \mathcal{A} \to \mathbb{R}$ is the (possibly uncertain) reward function assumed to be supported on a bounded interval $[0, R_{\max}]$. A stationary policy $\pi \in \Pi(\mathbf{M})$ is defined as $\pi : \mathcal{S} \to \Delta(\mathcal{A})$, meaning that at each discrete time step $t$, the agent observes a state $s_t$, samples an action $a_t \sim \pi(\cdot \mid s_t)$, collects a reward $r_t \sim R(\cdot \mid s_t, a_t)$, and transitions to $s_{t+1} \sim P(\cdot \mid s_t, a_t)$. We mention detailed notations used in this work in Table 3.

### 3.1. Value Functions and Policies

We adopt a discounted formulation with discount factor $\gamma \in (0, 1)$. The state-value and state–action value functions of a policy $\pi$ are given by

$$V_{P,R}^{\pi}(s) = \sum_{t=0}^{\infty} \mathbb{E}_{a_t \sim \pi}\left[\gamma^t \, r_t \mid s_0 = s\right], \qquad (1)$$

$$Q_{P,R}^{\pi}(s, a) = \sum_{t=0}^{\infty} \mathbb{E}_{a_t \sim \pi}\left[\gamma^t \, r_t \mid s_0 = s, \, a_0 = a\right]. \quad (2)$$

The *optimal* state-value function is defined as $V_{P,R}^{\star}(s) = \sup_{\pi} V_{P,R}^{\pi}(s)$. By definition and existence of deterministic optimal actions, the *optimal* state–action value function $Q^*$ satisfies $V_{P,R}^{\star}(s) = \max_a Q_{P,R}^{\star}(s, a)$ for each $s \in \mathcal{S}$.

### 3.2. Conceptualization of Robustness

A key challenge in real-world RL is that both transitions $P$ and rewards $R$ may be partially unknown or even time-varying. Let $P^o$ and $\nu^o$ denote the nominal transition probabilities and reward distributions, respectively, with each reward $r(s, a) \sim \nu_{s,a}^o$. These nominal models can be either factory-set approximations or a simulator of real-world systems. Following Wang et al. (2024b); Zhou et al. (2021); Liu et al. (2022), we allow the environment to deviate from $(P^o, \nu^o)$ within a *robustness budget* $\rho_{\mathrm{T}}, \rho_{\mathrm{R}}$ respectively. This leads to a robust MDP that accounts for uncertainties in both transitions *and* rewards.

**Ambiguity Sets.** We model transitions in an ambiguity set $\mathcal{P} = \bigotimes_{(s,a)} \mathcal{P}_{s,a}$, where each $\mathcal{P}_{s,a}$ contains all plausible distributions over next states from $(s, a)$. Analogously, an ambiguity set $\mathcal{R} = \bigotimes_{(s,a)} \mathcal{R}_{s,a}$ captures deviations in the reward distributions $r(s, a)$. Here, with a chosen metric

$D(\cdot, \cdot)$,

$$\mathcal{P}_{s,a} = \Big\{ P_{s,a} \in \Delta(\mathcal{S}) : \ D\big(P_{s,a}, P_{s,a}^o\big) \leq \rho_{\mathrm{T}} \Big\},$$

and

$$\mathcal{R}_{s,a} = \Big\{ \nu_{s,a} : \ D\big(\nu_{s,a}, \nu_{s,a}^o\big) \leq \rho_{\mathrm{R}} \Big\}.$$

Different choices of $D$ lead to distinct ambiguity sets, such as total-variation balls ($\mathcal{P}^{\mathrm{TV}}$), chi-squared neighborhoods ($\mathcal{P}^X$), or Wasserstein sets ($\mathcal{P}^W$). For notational convenience, we denote the reward distributions $\nu_{s,a} \in \mathcal{R}_{s,a}$ also as their probability densities in the context of measuring distances $D(\cdot, \cdot)$.

# 4. Main Problem Formulation

This section establishes how Monte Carlo Tree Search (MCTS) can be adapted to account for model ambiguity in a robust Markov Decision Process (MDP). Our goal is twofold: first, to clarify the root assumptions behind the robust planning framework, and second, to describe how MCTS is modified so that each node's value estimate incorporates worst-case rewards and transitions.

**Robust MDP.** We consider a *robust MDP* $\mathbf{M} = (\mathcal{S}, \mathcal{A}, \mathcal{P}, \mathcal{R})$ in which the state space $\mathcal{S}$ may be large or partially continuous, the action space $\mathcal{A}$ is discrete, and the unknown reward $r(s,a)$ and transition model $\mathcal{P}(\cdot \mid s,a)$ can lie within an *ambiguity set* $\mathcal{R}$ and $\mathcal{P}$ (described in Section 3). At each step $t$, the agent observes a state $s_t$, selects an action $a_t \in \mathcal{A}$, receives reward $r_t$, and transitions to a new state $s_{t+1}$. The robust state-value and state-action value functions of a policy $\pi$ are given by $V^\pi(s) = \min_{P \in \mathcal{P}, R \in \mathcal{R}} V_{P,R}^\pi(s)$ and $Q^\pi(s,a) = \min_{P \in \mathcal{P}, R \in \mathcal{R}} Q_{P,R}^\pi(s,a)$ respectively. A policy $\pi^\star$ that maximizes the value function is an optimal robust policy with corresponding optimal robust value functions $V^\star(s)$ and $Q^\star(s,a)$. Hence, both transitions and rewards may be adversarially perturbed, ensuring the agent plans robustly for worst-case scenarios within these sets.

**Robust Bellman Operator.** In the robust MDP, the *worst-case* expected value arises from an adversarial choice of both transition and reward distributions within their respective ambiguity sets. From the robust MDP literature (Iyengar, 2005; Liu et al., 2022), by the construction of $\mathcal{P}$ and $\mathcal{R}$ ambiguity sets, $Q^\star$ is known to be computable, and thereby $\pi^\star(s) = \mathrm{argmax}_{a \in \mathcal{A}} Q^\star(s,a)$.

Let us define for any set $\mathbf{B}$ and a vector $v$, $\sigma_{\mathbf{B}}(v) = \inf\{u^T v : u \in \mathcal{B}\}$. Robust dynamic programming, given by $V_{k+1}(s) = \max_{a \in \mathcal{A}} Q_{k+1}(s,a)$ and

$$Q_{k+1}(s,a) = R_{s,a}^{\mathrm{rob}} + \gamma \, \sigma_{\mathcal{P}_{s,a}}(V_k),$$

where $R_{s,a}^{\mathrm{rob}} = \min_{r_{s,a} \in \mathcal{R}_{s,a}} \mathbb{E}_{R \sim r_{s,a}}[R]$, and $\sigma_{\mathcal{P}_{s,a}}(V)$ captures the worst-case expected reward at $(s,a)$ and value of $V$ over $\mathcal{P}_{s,a}$, converges to optimal robust value functions $V^\star$ and $Q^\star$ respectively.

**MCTS in a Robust MDP.** In Monte Carlo Tree Search, we approximate a $\gamma$-discounted solution by simulating trajectories down a growing search tree. Each node corresponds to a state $s_h$, with $h$ indicating the depth in the tree (distance from the root). From $s_h$, the algorithm either expands a child node for the next state $s_{h+1}$ or performs a *rollout* using a simpler policy $\pi_0$ if $h$ reaches the maximum search depth $H$. Trajectories terminate upon reaching depth $H$ or a terminal state.

**Performance Measure.** A canonical metric for MCTS algorithms is the *convergence rate* $r(t)$, where $t$ indexes the number of simulated trajectories (rollouts). Informally, $r(t)$ bounds how quickly the MCTS estimates approach the true optimal values at the root node. For instance, one may require that $\mathbb{E}\big[ V^\star(s_0) - Q^\star(s_0, \widehat{a}_t) \big] \leq r(t)$, or

$$\big| \mathbb{E}\big[ V^\star(s_0) - \widehat{V}_t(s_0) \big] \big| \leq r(t),$$

where $\widehat{a}_t$ is the action chosen at the root after $t$ rollouts, and $\widehat{V}_t(s_0)$ approximates $V^\star(s_0)$.

**Recursive Value Estimation Under Ambiguity.** To capture the robust (worst-case) aspect of the MDP, we define a recursive estimation scheme at each node that accounts for $\inf_{r(s,a) \in \mathcal{R}_{s,a}}$ of reward and $\inf_{P \in \mathcal{P}_{s,a}}$ transitions. Let $s_h$ be a node at depth $h$. We assign a *robust value* $\widetilde{V}(s_h)$ and a *robust action-value* $\widetilde{Q}(s_h, a)$ such that

$$\widetilde{Q}(s_h, a) = R_{s,a}^{\mathrm{rob}} + \gamma \, \sigma_{\mathcal{P}_{s_h,a}}(\widetilde{V}),$$

$$\widetilde{V}(s_h) = \max_{a \in \mathcal{A}} \widetilde{Q}(s_h, a).$$

At a leaf node ($h = H$), we approximate the value with a simple rollout policy $\pi_0$, yielding $\widetilde{V}(s_H) \approx V_{\pi_0}(s_H)$.

**Goal of MCTS.** Since finite sample sizes introduce noise, each node's robust value $\widetilde{V}(s_h)$ is estimated from rollouts. The ultimate objective is to identify an action $a_\star = \arg\max_a Q^\star(s_0, a)$ at the root state $s_0$ within $n$ simulated trajectories, where $Q^\star(s_0, a)$ represents the robust-optimal action value. Intuitively, we want:

$$\widehat{a}_n \approx \arg\max_a \widetilde{Q}(s_0, a), \quad \widehat{V}_n(s_0) \approx \widetilde{V}(s_0),$$

with small statistical error. In Section 5, we describe how `Robust-Power-UCT` achieves this via specially designed backup operators and action-selection rules. Section 6 establishes finite-sample guarantees, showing that *robustness* in MCTS need not degrade convergence speed compared to its non-robust counterpart.

# 5. Algorithm Description

We now describe the core parts of our `Robust-Power-UCT` algorithm, focusing on the *value backup* and *action selection* strategies. Other details, such as the main loop and rollout procedure, are standard MCTS routines and hence only briefly mentioned.

Table 1: Key Conditions for Algorithmic Constants ($i \in [0, H]$)

| Cond. | Requirement |
|---|---|
| (1) | $b_i < \alpha_i$ and $b_i > 2$. |
| (2) | $\begin{cases} 1 \le p \le 2 \quad \text{and} \quad \alpha_i \le \frac{\beta_i}{2}, \\ \text{or} \\ p > 2 \quad \text{and} \quad \alpha_i \le \frac{\beta_i}{2},\, 0 < \alpha_i - \frac{\beta_i}{p} < 1 \end{cases}$ |
| (3) | $\alpha_i \left(1 - \frac{b_i}{\alpha_i}\right) \le b_i < \alpha_i$. |
| (4) | $\alpha_i = (b_{i+1} - 1)\left(1 - \frac{b_{i+1}}{\alpha_{i+1}}\right)$. |
| (5) | $\beta_i = (b_{i+1} - 1)$. |

**Value Backup.** To estimate the value function at each node, we use a *power mean* backup operator. When node $s_h$ is expanded in the tree, we define inductively for all $t$,

$$\widehat{V}_t(s_h) = \left( \sum_{a \in \mathcal{A}_{s_h}} \frac{T_{s_h,a}(t)}{t} \left[ \widehat{Q}_{T_{s_h,a}(t)}(s_h, a) \right]^p \right)^{\frac{1}{p}},$$

where $p \ge 1$. This *power mean* backup places more emphasis on actions that have high current value estimates (when $p > 1$), but still captures the contributions of other actions. Meanwhile $\widehat{Q}_{T_{s_h,a}(t)}(s_h, a)$, or simply $\widehat{Q}_t(s_h, a)$ as the root is $s_h$, itself is updated via

$$\widehat{Q}_t(s_h, a) = \widehat{R}_{s_h,a}^{\text{rob}} + \gamma \sigma_{\widehat{\mathcal{P}}_{s_h,a}}\big(\widehat{V}_{T_{s_{h+1}}(t)}\big), \quad (3)$$

where $\widehat{R}_{s_h,a}^{\text{rob}} = \min_{r \in \widehat{\mathcal{R}}_{s_h,a}} \mathbb{E}_{R \sim r}[R]$ is an empirical robust reward at $(s_h, a)$, and $\sigma_{\widehat{\mathcal{P}}}(\cdot)$ is a *robust operator* capturing worst-case transitions for ambiguity sets governed by empirical estimates of nominal reward and transition models:

$$\widehat{\mathcal{P}}_{s,a} = \Big\{ P_{s,a} \in \Delta(\mathcal{S}) \colon D\big(P_{s,a}, \widehat{p}_{s,a}\big) \le \rho_{\text{T}} \Big\},$$

and

$$\widehat{\mathcal{R}}_{s,a} = \Big\{ \nu_{s,a} \in \Delta(B) \colon D\big(\nu_{s,a}, \widehat{\nu}_{s,a}\big) \le \rho_{\text{R}} \Big\}.$$

**Action Selection.** At each node $s_h$ in the search tree, `Robust-Power-UCT` selects an action $a$ according to an *optimistic* rule that balances exploration and exploitation.

Specifically, we maintain an empirical estimate $\widehat{Q}_t(s_h, a)$ for each action and add an exploration bonus of the form:

$$C \cdot \frac{\big(T_{s_h}(t)\big)^{\frac{b_{h+1}}{\beta_{h+1}}}}{\big(T_{s_h,a}(t)\big)^{\frac{\alpha_{h+1}}{\beta_{h+1}}}},$$

where $T_{s_h}(t)$ is the total number of visits to $s_h$ up to time $t$, and $T_{s_h,a}(t)$ is how often action $a$ has been taken from $s_h$. The exponents $\frac{b_{h+1}}{\beta_{h+1}}$ and $\frac{\alpha_{h+1}}{\beta_{h+1}}$ control how aggressively the algorithm explores, while $C$ is a user-chosen constant. At the end of training (greedy mode), the action with the highest $\widehat{Q}_t$ is chosen.

**Main Loop and Rollout.** As in standard MCTS, the algorithm repeatedly simulates from the root state $s_0$, selecting actions according to the above scheme. When reaching a leaf node (unexpanded or maximum depth), a *rollout policy* approximates the return from that leaf. These routines are routine and can be implemented similarly to classical MCTS methods.

By combining an *optimistic action selection* mechanism with a *power mean* and robust operator for value backup, `Robust-Power-UCT` systematically balances exploration of uncertain actions and exploitation of promising ones, all under model ambiguity.

# 6. Theoretical Results

In robust MCTS planning, each internal node of the search tree can be viewed as a *non-stationary* multi-armed bandit due to ongoing updates of the node's reward and transition ambiguity estimates. At each step, the empirical evaluations shift, reflecting how robust exploration is balanced against uncertainty in the model. To handle this dynamic process, we begin by studying a non-stationary multi-armed bandit problem—focusing on how the power-mean backup operator concentrates around its robust-optimal value. We then leverage these results to prove convergence properties of our robust MCTS algorithm, showing that it systematically discards suboptimal branches under model uncertainty while maintaining sample efficiency.

## 6.1. Non-Stationary Bandit Perspective

We first analyze `Robust-Power-UCT` in a simpler non-stationary multi-armed bandit setting. Here, actions are selected optimistically, and the *power mean backup* operator is used at the root node.

### 6.1.1. PROBLEM DESCRIPTION AND KEY DEFINITIONS

We consider a class of *non-stationary multi-armed bandit* (MAB) problems with $K \ge 1$ actions (arms) with the reward $\in [0, R]$. Define a sequence of estimator $\widehat{\mu}_{a,n}$ (in this

---

Algorithm 1: `Robust-Power-UCT` with $\gamma$ discount factor. $n$ : the number of rollouts. $\{b_i, \alpha_i, \beta_i\}_{i=0}^H$ are positive constants that satisfy conditions in Table 1. $B$ is the total bins for $[0, R_{\max}]$. $\pi_0$ is a rollout policy. $C$ is an exploration constant.

**Input:** root node state $s_0$
**Output:** optimal action at the root node

**Function** $R$ = Rollout $(s, depth)$
  $\widetilde{V}(s)$ = average of the call to $\pi_0(s)$
  **return** $\widetilde{V}(s)$

**Function** $a$ = SelectAction $(s_h, depth = h, greedy = false, t)$
  **if** $greedy$ == *false* **then**
    $a = \underset{a}{\operatorname{argmax}}\{\widehat{Q}_{T_{s_h,a}(t)}(s_h, a) + C \frac{T_{s_h}(t)^{\frac{b_{h+1}}{\beta_{h+1}}}}{T_{s_h,a}(t)^{\frac{\alpha_{h+1}}{\beta_{h+1}}}}\}$
  **else**
    $a = \underset{a}{\operatorname{argmax}}\{\widehat{Q}_{T_{s_h,a}(t)}(s_h, a)\}$
  **end**
  **return** $a$

**Function** SimulateV $(s_h, depth, t)$
  $a \leftarrow$ SelectAction$(s_h, depth = h, greedy = false, t)$
  SimulateQ $(s_h, a, depth = h, t)$
  $T_{s_h}(t) \leftarrow T_{s_h}(t) + 1$
  $\widehat{V}_{T_{s_h}(t)}(s_h) \leftarrow \left(\sum_a \frac{T_{s_h,a}(t)}{T_{s_h}(t)}(\widehat{Q}_{T_{s_h,a}(t)}(s_h, a))^p\right)^{\frac{1}{p}}$

**Function** SimulateQ $(s_h, a, depth = h, t)$
  $s_{h+1} \sim P^o(\cdot|s_h, a)$
  $r(s_h, a) \sim \nu^o_{s_h,a}$
  **if** $s_{h+1} \notin Terminal$ and $depth \leqslant H - 1$ **then**
    **if** *Node* $s_{h+1}$ *not expanded* **then**
      $\widehat{V}_{T_{s_{h+1}}(t)}(s_{h+1}) =$ Rollout$(s_{h+1}, depth)$
    **else**
      SimulateV $(s_{h+1}, depth = h + 1, t)$
    **end**
  **end**
  **Find** $j \in B$ *s.t.* $r(s_h, a) \in Bin^j[0, R_{\max}]$
    $\widehat{\nu}_{s_h,a}(j) = \frac{\widehat{\nu}_{s_h,a}(j) \cdot T_{s_h,a}(t) + 1}{T_{s_h,a}(t) + 1}$
  $\widehat{p}_{s_h,a}(s_{h+1}) = \frac{\widehat{p}_{s_h,a}(s_{h+1}) \cdot T_{s_h,a}(t) + 1}{T_{s_h,a}(t) + 1}$
  $T_{s_h,a}(t) \leftarrow T_{s_h,a}(t) + 1$
  $\widehat{Q}_{T_{s_h,a}(t)}(s_h, a) \leftarrow \widehat{R}^{\text{rob}}_{s_h,a} + \gamma \sigma_{\widehat{p}_{s_h,a}}(\widehat{V}_{T_{s_{h+1}}(t)})$

**Function** MainLoop
  **For** $t = 0, \cdots, n$
    SimulateV $(s_0, depth = 0, t)$
  **return** SelectAction $(s_0, greedy = true, n)$

---

paper is the robustness estimation of the mean value of arm $a$) such that $\mu_{a,n} = \mathbb{E}[\widehat{\mu}_{a,n}]$. We are interested in sequence of estimators that satisfy a suitable concentration property:

**Definition 1** (Concentration). *A sequence of estimators* $\{\widehat{Y}_n\}_{n \geq 1}$ *concentrates at rate* $(\alpha, \beta)$ *toward a limit* $Y$*, writing as* $\widehat{Y}_n \overset{\alpha,\beta}{\underset{n\to\infty}{\to}} Y$*, if there is a constant* $c > 0$ *such that*

$$\forall n \geq 1, \ \forall \varepsilon > n^{-\frac{\alpha}{\beta}}, \ \mathbf{Pr}\left(|\widehat{Y}_n - Y| > \varepsilon\right) \leq c \, n^{-\alpha} \varepsilon^{-\beta}.$$

**Assumption 1** (Non-Stationary Rewards). *For each arm* $a \in [K]$*, the sequence* $\{\widehat{\mu}_{a,n}\}_{n \geq 1}$ *concentrates at rate* $(\alpha, \beta)$ *toward a value* $\mu_a$*, i.e.* $\widehat{\mu}_{a,n} \overset{\alpha,\beta}{\underset{n\to\infty}{\to}} \mu_a$*. Let* $\mu_\star = \max_{a \in [K]}\{\mu_a\}$*, assumed to be unique with a strict gap from suboptimal* $\mu_a$*.*

### 6.1.2. OPTIMISTIC ACTION SELECTION AND POWER MEAN BACKUP

Under Assumption 1, we use an *optimistic* exploration rule similar to `Robust-Power-UCT`. Let $T_a(n)$ be the number of times arm $a$ is pulled before time $n$. The algorithm pulls each arm once initially. For $n > K$:

$$a_n = \arg \max_{a \in [K]}\left\{\widehat{\mu}_{a,T_a(n)} + C n^{\frac{b}{\beta}}/T_a(n)^{\frac{\alpha}{\beta}}\right\}, \quad (4)$$

where $b > 2$ and $b < \alpha$. For the power mean operator, let $p \in [1, \infty)$ and define $\widehat{\mu}_n(p) =$

$\left(\sum_{a=1}^K \frac{T_a(n)}{n}\left[\widehat{\mu}_{a,T_a(n)}\right]^p\right)^{\frac{1}{p}}$. By applying Theorem 1 of Dam et al. (2024b), we get $\widehat{\mu}_n(p) \overset{\alpha',\beta'}{\underset{n\to\infty}{\to}} \mu_\star$, where $\alpha' = (b-1)\left(1 - \frac{b}{\alpha}\right)$, and $\beta' = (b-1)$.

**Connecting Back to MCTS.** This bandit analysis underpins how `Robust-Power-UCT` handles exploration and the power mean backup. In an MCTS context, each node's local bandit analysis is augmented by worst-case backups, but the principle is similar: the algorithm discards suboptimal branches with high probability, causing the robust estimates to concentrate around the best actions.

### 6.1.3. MAIN CONVERGENCE RESULTS

Before presenting the main result (Theorem 3), we first show an important lemma used for our MCTS algorithm.

**Lemma 17.** *For* $m \in [M]$*, let* $(\widehat{V}_{m,n})_{n \geqslant 1}$ *be a sequence of estimator satisfying* $\widehat{V}_{m,n} \overset{\alpha,\beta}{\underset{n\to\infty}{\to}} V_m$*, and there exists a constant* $L$ *such that* $\widehat{V}_{m,n} \leqslant L, \forall n \geqslant 1$*. Let* $X_i$ *be an iid sequence from a distribution* $\nu^o$ *with mean* $\mu$ *and* $S_i$ *be an iid sequence from a distribution* $p = (p_1, \ldots, p_M)$ *supported on* $\{1, \ldots, M\}$*. Introducing the random variables* $N_m^n = \#|\{i \leqslant n : S_i = s_m\}|$*. Define a model estimate of* $p$ *as* $\widehat{p}_n = (\frac{N_1^n}{n}, \frac{N_2^n}{n}, ..., \frac{N_M^n}{n})$*. We define an estimate of* $\nu^o$ *as* $\widehat{\nu}_n = \frac{1}{n}\sum_{i=1}^n \delta_{X_i}$*. Recall* $R^{\text{rob}} = \min_{r \in \mathcal{R}} \mathbb{E}_{R \sim r}[R]$ *w.r.t* $\nu^o$ *and* $\widehat{R}^{\text{rob}} = \min_{r \in \widehat{\mathcal{R}}} \mathbb{E}_{R \sim r}[R]$ *w.r.t* $\widehat{\nu}_n$*. We define*

the sequence of estimators

$$\widehat{Q}_n = \widehat{R}^{\mathrm{rob}} + \gamma \sigma_{\widehat{p}_n}(\widehat{V}_n).$$

Then with $2\alpha \leqslant \beta, \beta > 1$, $\widehat{Q}_n \overset{\alpha,\beta}{\underset{n\to\infty}{\to}} R^{\mathrm{rob}} + \gamma\sigma_p(V)$.

**Remark 1.** *This non-asymptotic convergence result shows that, for suitable parameters $(\alpha, \beta)$, the estimator $\widehat{Q}_n$ will concentrate around the limiting quantity $R^{\mathrm{rob}} + \gamma\,\sigma_p(V)$ with high probability. Importantly, we do not claim these $(\alpha, \beta)$ are in any sense optimal; rather, we only need the existence of such parameters that guarantee the concentration at the prescribed rate. Moreover, our analysis uses covering number generalization to handle continuous reward distributions. Furthermore, the constant $c$ implicit in the notation $\widehat{V}_n \overset{\alpha,\beta}{\underset{n\to\infty}{\to}} V$ can depend on problem-dependent factors (e.g., size of the action set $A$, number of states $S$, etc.), reflecting the stochastic process complexity.*

### 6.2. Tree-Level Convergence

The above non-stationary bandit analysis is critical for proving the subsequent tree-level theorems. In particular, Theorem 2 (restated below) shows that under appropriate parameter settings (Table 1), the estimated node values $\widehat{V}_n(\cdot)$ and $\widehat{Q}_n(\cdot, \cdot)$ converge at a known rate:

**Theorem 2.** *When applying* `Robust-Power-UCT` *with parameters $\{b_i\}_{i=0}^H$, $\{\alpha_i\}_{i=0}^H$, $\{\beta_i\}_{i=0}^H$ satisfying Table 1:*

*(i) For any node $s_h$ at depth $h \in \{0, \ldots, H\}$,*

$$\widehat{V}_n(s_h) \overset{\alpha_h,\beta_h}{\underset{n\to\infty}{\to}} \widetilde{V}(s_h).$$

*(ii) For any node $s_h$ at depth $h \in \{0, \ldots, H-1\}$,*

$$\widehat{Q}_n(s_h, a) \overset{\alpha_{h+1},\beta_{h+1}}{\underset{n\to\infty}{\to}} \widetilde{Q}(s_h, a), \quad \forall a \in \mathcal{A}_{s_h}.$$

*Proof.* (Sketch) The argument proceeds by induction on the tree depth $H$. For $H = 1$, we handle the root node using Lemma 17 plus the concentration assumptions on leaf nodes. For general $H$, we note that descending into a child node effectively reduces the depth by one, thus the induction hypothesis applies. By carefully controlling exploration (Section 5) and using robust backups, each node's $\widehat{V}$ and $\widehat{Q}$ estimates concentrate at the specified rates. □

Finally, Theorem 3 establishes that under optimal parameter tuning, the expected payoff at the root converges at $\mathcal{O}(n^{-1/2})$.

**Theorem 3.** *(Convergence of Expected Payoff) At the root node $s_0$, there is a choice of parameters yielding*

$$\left| \mathbb{E}\big[\widehat{V}_n(s_0)\big] - \widetilde{V}(s_0) \right| \leq \mathcal{O}\big(n^{-1/2}\big).$$

**Remark 2.** *These results show that both* `Robust-Power-UCT` *and standard (non-robust) MCTS achieve the same $\mathcal{O}(n^{-1/2})$ rate for value estimation at the root node, which implies that robustness need not affect convergence speed, which is order-optimal. While we achieve this rate, the exact dependence on various problem-dependent factors (e.g., number of actions $A$, number of states $S$, tree search depth $H$, etc.) is not decodable (thereby not comparable to other online robust RL results (Dong et al., 2022)) due to our analysis limitations.*

## 7. Experiments

We evaluate `Robust-Power-UCT` in three distinct environments designed to test different aspects of robust planning: the Gambler's Problem, Frozen Lake, and American Option Pricing. For each environment, we compare: Stochastic-Power-UCT (Dam et al., 2024b) (baseline) and Robust-Power-UCT with Total Variation, Chi-squared, and Wasserstein ambiguity sets.

While several robust reinforcement learning methods exist (c.f.Section 2), to the best of our knowledge, this is the first work to incorporate ambiguity sets directly into MCTS, making Stochastic-Power-UCT our primary baseline. All experiments are done over 100 seeds, using $\gamma = 0.99$ and robustness budget $\rho = 0.5$, with these values showing consistent performance across preliminary experiments with different parameter settings. For concise presentation, we only experiment with transition model ambiguity just as prior robust RL works.

Full experimental details, environment descriptions and hyperparameter configurations are provided in Appendix.E.1, along with an additional analysis of the robustness budget. We also provide our code at `https://github.com/brahimdriss/RobustMCTS`.

**Remark 3.** *While the robust Bellman operator involves solving a minimization problem over probability distributions, we can leverage dual reformulations to make its computation tractable. Many prior works (Iyengar, 2005; Nilim and El Ghaoui, 2005; Xu et al., 2023) show, for a value function $V$ and nominal distribution $P^o$, the robust value under all ambiguity balls with radius $\rho_T$ can be computed in at most $\mathcal{O}(S\log(S))$ time. Thus requiring only marginally more computation than standard Bellman operators $\mathcal{O}(S)$. This computational efficiency is crucial for practical implementations, particularly in online planning settings like MCTS with frequent Bellman updates.*

### 7.1. Gambler's Problem Robustness Results

The Gambler's Problem provides an ideal testbed for evaluating robustness to model misspecification. An agent must reach a target capital through a series of bets, with each

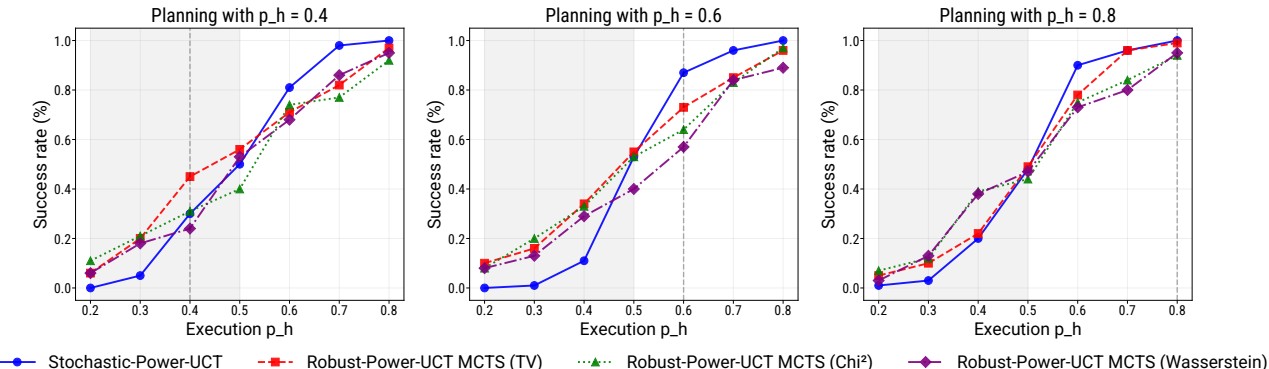

Figure 1: Success rates in the Gambler's Problem under model mismatch. Results show planning with fixed probabilities $p_h = \{0.4, 0.6, 0.8\}$ while executing across different probabilities. Shaded area demonstrates how robust methods maintain more consistent performance under model mismatch compared to Stochastic-Power-UCT.

bet winning with probability $p_h$. This enables precise control of the planning-execution mismatch through a single parameter.

Figure 1 illustrates the performance of different Power-UCT variants under model mismatch in the Gambler's Problem. The behavior of Stochastic-Power-UCT reveals a fundamental vulnerability: when $p_h < 0.5$, there exist multiple optimal policies that achieve winning ratios close to the true environment probability. However, when planning with $p_h \geqslant 0.5$, the algorithm converges to an aggressive single-bet strategy that fails catastrophically when the true probability is lower than assumed.

The superior performance of robust variants stems from their conservative betting strategies. While Stochastic-Power-UCT often makes large single bets, robust variants tend to make smaller, sequential bets that preserve capital for future opportunities.

### 7.2. Frozen Lake Robustness Results

The Frozen Lake environment tests robustness in a different complex setting where uncertainties compound over multiple steps. The agent must navigate to a goal while avoiding hazards, with actions potentially failing with probability $p_{\text{slip}}$.

Table 2 provides detailed success rates across different planning and execution probabilities. With matching conditions (4000 rollouts and $p_{\text{slip}} = 0.3$ case), our results closely match those reported in the original paper (Dam et al., 2024b) even with slightly different dynamics. The Wasserstein uncertainty set exhibits superior performance in scenarios with lower execution probabilities, achieving the highest success rates (bold) across multiple conditions. For example, with $p_{\text{slip}} = 0.3$, it achieves 58% success when $p_{\text{exec}} = 0.1$, significantly outperforming other ap-

proaches.

Both Wasserstein and Chi-squared variants outperform the baseline Stochastic-Power-UCT and Total Variation approaches. Interestingly, when planning and execution probabilities align (underlined values), both robust variants maintain superior performance compared to standard approaches. This suggests that explicitly accounting for uncertainty in the planning process provides benefits even without model mismatch, possible by encouraging more conservative and reliable decision-making strategies.

These results on Gambler's Problem and Frozen Lake demonstrate that explicitly accounting for model ambiguity during planning can significantly improve reliability when deployment conditions differ from simulation assumptions. The choice of ambiguity set provides a mechanism for balancing conservatism against nominal performance.

### 7.3. American Option Robustness Results

The American Option environment provides a financial domain to test reward robustness under model uncertainty. In this setting, the agent must decide when to exercise an option to maximize expected returns, with the key uncertain parameter being the probability $p_u$ of price increases at each time step.

Figure 2 demonstrates the reward robustness of different Power-UCT variants under model mismatch in option pricing scenarios. We examine two planning scenarios: training with $p_u = 0.5$ (left panel) and $p_u = 0.6$ (right panel), then testing across execution probabilities from 0.4 to 0.8.

The results reveal that robust variants maintain significantly more stable performance compared to standard Power-UCT. When planning with $p_u = 0.5$, the standard approach shows dramatic performance degradation as the test probability deviates from the planning assumption, dropping

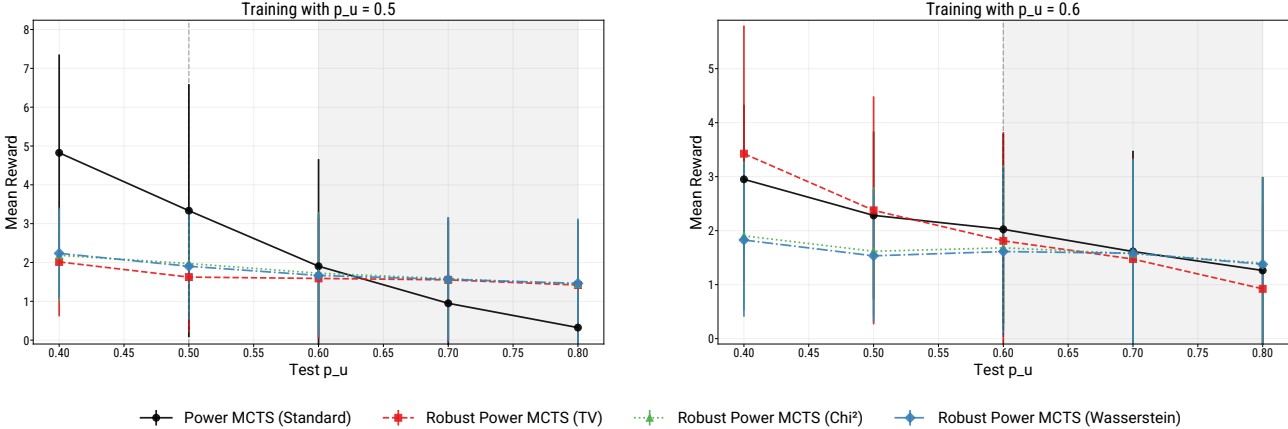

Figure 2: Reward robustness comparison in American Option pricing under model mismatch. Results show planning with fixed price-up probabilities $p_u = \{0.5, 0.6\}$ while testing across different probabilities. Robust variants maintain significantly more stable performance compared to standard Power-UCT, demonstrating consistent risk-averse behavior that is particularly valuable in financial decision-making contexts where reliability is crucial.

| Planning | | Execution $p_{\text{slip}}$ | | | | |
|---|---|---|---|---|---|---|
| $p_{\text{slip}}^{\text{plan}}$ | | 0.1 | 0.2 | 0.3 | 0.4 | 0.5 |
| | Sp | 15 | 12 | 10 | 8 | 7 |
| 0.3 | Tv | 18 | 15 | 12 | 10 | 8 |
| | Cs | 55 | 45 | **35** | 25 | 18 |
| | Ws | **58** | **48** | 32 | **28** | **20** |
| | Sp | 8 | 7 | 6 | 5 | 4 |
| 0.4 | Tv | 10 | 8 | 7 | 6 | 5 |
| | Cs | 35 | 28 | 22 | 18 | 12 |
| | Ws | **38** | **30** | **25** | **20** | **15** |
| | Sp | 5 | 4 | 4 | 3 | 3 |
| 0.5 | Tv | 6 | 5 | 4 | 4 | 3 |
| | Cs | 25 | 20 | 15 | 12 | 8 |
| | Ws | **28** | **22** | **18** | **15** | **10** |

Table 2: Success rates (%) for planning with Power-UCT variants. Methods: Stochastic-Power-UCT (Sp), Robust version with Total Variation (Tv), Chi-squared (Cs), and Wasserstein (Ws) ambiguity sets. Underlined values indicate matching planning and execution $p_{\text{slip}}$. Bold indicates highest success rate per planning scenario.

from approximately 5 to near 0 when $p_u = 0.8$. In contrast, robust variants maintain consistent performance across the entire range.

When planning with $p_u = 0.6$, standard Power-UCT exhibits extreme sensitivity with dramatically varying performance. The robust variants demonstrate desired risk-averse behavior: achieving conservative but stable returns across all conditions. This stability is especially valuable in fi-

nancial contexts where consistent performance is preferred over potentially high but unreliable returns.

The Wasserstein and Chi-squared ambiguity sets show particularly strong performance, maintaining steady rewards even under significant model mismatch, demonstrating that explicitly accounting for uncertainty leads to policies inherently more robust to different deployment conditions.

## 8. Conclusions

We have developed a robust variant of Monte Carlo Tree Search (MCTS) that addresses dynamical model and reward distribution ambiguities, bridging the gap between simulation-based planning and real-world deployment. The dependence of MCTS-based algorithms' convergence rates on parameters (states $S$, actions $A$, depth $H$) remains underexplored in standard RL. We will address this gap for both robust and non-robust setups in the future. As our formulation follows an overly conservative mathematical framework, in the future, we will explore alternative robust formulations that are more permeable to less conservative solutions to address the simulation-to-reality gap.

## Impact Statement

This paper presents a novel algorithm for the robust reinforcement learning field using the Monte Carlo Tree Search planning mechanism. There are many potential societal consequences of our work, none of which we feel must be specifically highlighted here.

## Acknowledgments

Tuan Dam was funded by Hanoi University of Science and Technology (HUST) under Project No. T2024-TD-024. K. Panaganti acknowledges support from the Resnick Institute and the 'PIMCO Postdoctoral Fellow in Data Science' fellowship at Caltech. B. Driss was funded by the project ANR-23-CE23-0006. A. Wierman acknowledges support by the NSF through CNS-2146814, CPS-2136197, CNS-2106403, and NGSDI-2105648. This work was granted access to the HPC resources of IDRIS under the allocation 2024-AD011015599 made by GENCI.

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

# A. Notations

| Notation | Description |
|---|---|
| $\mathcal{S}, \mathcal{A}$ | State space and action space of the MDP. |
| $H$ | Planning horizon (depth of the search tree). |
| $(s, a)$ | A specific state–action pair; $s \in \mathcal{S}, a \in \mathcal{A}$. |
| $[M]$ | Denotes the set $\{1, 2, \cdots, M\}$. |
| $\nu_{s,a}^o$ | Nominal (true) reward distribution for state–action pair $(s, a)$. |
| $\nu_{s,a}, \widehat{\nu}_{s,a}$ | Generic and empirical reward distributions for $(s, a)$, respectively. |
| $R_{\max}$ | Maximum possible reward value (i.e., reward is supported in $[0, R_{\max}]$). |
| $D_f(P \| P^o)$ | $f$-divergence between distributions $P$ and $P^o$, for a convex $f(\cdot)$. |
| $\mathcal{P}_{s,a}^{TV}, \mathcal{P}_{s,a}^{\mathcal{X}}, \mathcal{P}_{s,a}^{\mathcal{W}}$ | Uncertainty sets under Total Variation, Chi-square, and Wasserstein distances, respectively. |
| $\sigma_{\mathcal{P}_{s,a}}(V)$ | Worst-case *value operator* (or "robust backup") over an uncertainty set $\mathcal{P}_{s,a}$. |
| $\rho$ | Radius (budget) for the uncertainty set in $f$-divergence or Wasserstein distance. |
| $\sigma_{\widehat{p}_n}(\widehat{V}_n)$ | Robust backup operator with the empirical transition $\widehat{p}_n$ as the set center for empirical value $\widehat{V}_n$. |
| $B_p$ | Constant bounding the metric space for Wasserstein distance (e.g. max distance $d^p$). |
| $\alpha^*, \widehat{\alpha}^*$ | Dual variables optimizing robust reward functions under TV uncertainty sets. |
| $\Delta(X)$ | Probability simplex over the support of set $X$ or size of $X$. |
| $\mathcal{N}_R(\theta)$ | $\theta$-cover set used for bounding the supremum of $(\eta - R)_+$ in total-variation analysis. |
| $\|\cdot\|_\infty, \|\cdot\|_1$ | Infinity norm (maximum absolute value in a vector) and $\ell_1$ norm (sum of absolute values in a vector). |
| $\delta_x$ | A point mass at a realization $x$. |
| $\delta, \theta, \varepsilon$ | Parameters often controlling confidence levels or approximation accuracy in concentration bounds. |
| $c, C$ | Constants from generic concentration or covering-number arguments (possibly problem-dependent). |
| $p_m, N_m^n$ | Used for i.i.d. sampling from a discrete distribution $(p_1, \ldots, p_M)$, with $N_m^n$ the count of outcomes of type $m$. |
| $\widehat{Q}_n, Q^\star$ | Estimated and true robust $Q$-values, respectively. |
| $\widehat{V}_n, V^\star$ | Estimated and true robust value functions, respectively. |
| $\gamma$ | Discount factor in the MDP. |
| $\mathcal{W}(\mu_n, \mu)$ | Wasserstein distance between empirical measure $\mu_n$ and true measure $\mu$. |
| $\overset{\alpha, \beta}{\underset{n \to \infty}{\longrightarrow}}$ | Notation for *concentration* at rate $(\alpha, \beta)$; see text for precise definition. |

Table 3: Key Notations Used in the Appendix. Symbols and definitions for uncertainty sets (TV, $\chi^2$, Wasserstein), reward distributions, and the main variables in robust MDP analysis.

# B. Useful technical results

**Lemma 1.** *(Lemma 1 (Panaganti and Kalathil, 2022)) For any $(s, a) \in \mathcal{S} \times \mathcal{A}$ and for any $V_1, V_2 \in \mathbb{P}^{|S|}$, we have* $|\sigma_{P_{s,a}}(V_1) - \sigma_{P_{s,a}}(V_2)| \leqslant \|V_1 - V_2\|_\infty$ *and* $|\sigma_{\widehat{P}_{s,a}}(V_1) - \sigma_{\widehat{P}_{s,a}}(V_2)| \leqslant \|V_1 - V_2\|_\infty$

**Lemma 2.** *(Proposition 2 (Xu et al., 2023)) Fix any $h, s, a \in [H] \times S \times A$. For any $\theta, \delta \in (0, 1)$, we have with the probability of at least $1 - \delta$,* $\left|\sigma_{P_{s,a}^{TV}}(\widehat{V}_{h+1}) - \sigma_{\widehat{P}_{s,a}^{TV}}(\widehat{V}_{h+1})\right| \leqslant 2\theta + \sqrt{H^2 \log(4H/\theta\delta)/2n}$

From Lemma 2, we have

$$\mathbb{P}\left(\left|\sigma_{P_{s,a}^{TV}}(\widehat{V}_{h+1}) - \sigma_{\widehat{P}_{s,a}^{TV}}(\widehat{V}_{h+1})\right| \geqslant 2\theta + \sqrt{H^2 \log(4H/\theta\delta)/2n}\right) < \delta \tag{5}$$

Set $\theta = \varepsilon/4$ with $\varepsilon = 2\sqrt{H^2 \log(4H/\theta\delta)/2n}$, then

$$2n\varepsilon^2/4 = H^2 \log(4H/\theta\delta) \Rightarrow \exp\{-2n\varepsilon^2/H^2\} = \varepsilon\delta/16H \tag{6}$$

$$\Rightarrow \delta = \frac{16H \exp\{-n\varepsilon^2/2H^2\}}{\varepsilon} \tag{7}$$

so that

$$\mathbb{P}\left(\left|\sigma_{P_{s,a}^{TV}}(\widehat{V}_{h+1}) - \sigma_{\widehat{P}_{s,a}^{TV}}(\widehat{V}_{h+1})\right| \geqslant \varepsilon\right) < \frac{16H \exp\{-n\varepsilon^2/2H^2\}}{\varepsilon} \tag{8}$$

**Lemma 3.** *(Proposition 4 (Xu et al., 2023)) Fix any $h, s, a \in [H] \times S \times A$. For any $\theta, \delta \in (0, 1)$, we have with the probability of at least $1 - \delta$,* $\left|\sigma_{P_{s,a}^{\mathcal{X}}}(\widehat{V}_{h+1}) - \sigma_{\widehat{P}_{s,a}^{\mathcal{X}}}(\widehat{V}_{h+1})\right| \leqslant 2\theta + \frac{\sqrt{2}C_p^2 H}{(C_p-1)\sqrt{n}}\left(\sqrt{\log\left(\frac{2(1+C_p H/(\theta(C_p-1)))}{\delta}\right)} + 1\right)$

Then we have

$$\mathbb{P}\left(\left|\sigma_{P_{s,a}^{\mathcal{X}}}(\widehat{V}_{h+1}) - \sigma_{\widehat{P}_{s,a}^{\mathcal{X}}}(\widehat{V}_{h+1})\right| \geqslant 2\theta \quad + \frac{\sqrt{2}C_p^2 H}{(C_p - 1)\sqrt{n}}\left(\sqrt{\log\left(\frac{2(1 + C_p H/(\theta(C_p - 1)))}{\delta}\right)} + 1\right)\right) < \delta \quad (9)$$

Set $\theta = \varepsilon/4$ with $\varepsilon = 2\frac{\sqrt{2}C_p^2 H}{(C_p - 1)\sqrt{n}}\left(\sqrt{\log\left(\frac{2(1 + C_p H/(\theta(C_p - 1)))}{\delta}\right)} + 1\right)$, then

$$\left(\frac{(C_p - 1)\sqrt{n}\varepsilon}{2\sqrt{2}C_p^2 H} - 1\right)^2 = \log\left(\frac{2(1 + C_p H/(\theta(C_p - 1)))}{\delta}\right) \quad (10)$$

$$\Rightarrow \exp\left\{-\left(\frac{(C_p - 1)\sqrt{n}\varepsilon}{2\sqrt{2}C_p^2 H} - 1\right)^2\right\} = \frac{\delta}{2(1 + C_p H/(\theta(C_p - 1)))} \quad (11)$$

$$\Rightarrow \mathbb{P}\left(\left|\sigma_{P_{s,a}^{\mathcal{X}}}(\widehat{V}_{h+1}) - \sigma_{\widehat{P}_{s,a}^{\mathcal{X}}}(\widehat{V}_{h+1})\right| \geqslant \varepsilon\right) < \quad (12)$$

$$2\left(1 + C_p H/\left(\frac{\varepsilon(C_p - 1)}{4}\right)\right)\exp\left\{-\left(\frac{(C_p - 1)\sqrt{n}\varepsilon}{2\sqrt{2}C_p^2 H} - 1\right)^2\right\} \quad (13)$$

**Lemma 4.** *(Proposition 9 (Xu et al., 2023)) Fix any $h, s, a \in [H] \times S \times A$. For any $\theta, \delta \in (0, 1)$, we have with the probability of at least $1 - \delta$, $\left|\sigma_{P_{s,a}^{\mathcal{W}}}(\widehat{V}_{h+1}) - \sigma_{\widehat{P}_{s,a}^{\mathcal{W}}}(\widehat{V}_{h+1})\right| \leqslant 2\theta + \frac{H(B_p + \rho^p)}{\rho^p}\sqrt{\frac{\log\left(\frac{2HB_p + 2H\sqrt{\rho^p}}{\rho^p\theta\delta}\right)}{2n}}$*

Similarly, Set $\theta = \varepsilon/4$ with $\varepsilon = \frac{2H(B_p + \rho^p)}{\rho^p}\sqrt{\frac{\log\left(\frac{2HB_p + 2H\sqrt{\rho^p}}{\rho^p\theta\delta}\right)}{2n}}$, then

$$\exp\left\{-\frac{2n\varepsilon^2\rho^{2p}}{4H^2(B_p + \rho^p)^2}\right\} = \frac{\rho^p\theta\delta}{2HB_p + 2H\sqrt{\rho^p}} \quad (14)$$

$$\Rightarrow \delta = \frac{4(2HB_p + 2H\sqrt{\rho^p})}{\rho^p\varepsilon}\exp\left\{-\frac{2n\varepsilon^2\rho^{2p}}{4H^2(B_p + \rho^p)^2}\right\} \quad (15)$$

so that

$$\mathbb{P}\left(\left|\sigma_{P_{s,a}^{\mathcal{W}}}(\widehat{V}_{h+1}) - \sigma_{\widehat{P}_{s,a}^{\mathcal{W}}}(\widehat{V}_{h+1})\right| \geqslant \varepsilon\right) < \frac{4(2HB_p + 2H\sqrt{\rho^p})}{\rho^p\varepsilon}\exp\left\{-\frac{2n\varepsilon^2\rho^{2p}}{4H^2(B_p + \rho^p)^2}\right\} \quad (16)$$

**Lemma 5.** *(Lemma 2 (Fournier and Guillin, 2015), Concentration inequality for Wasserstein distance ). For $\mu \in \mathcal{P}(\mathbb{R})$, we consider an i.i.d. sequence $(X_k)_{k \geqslant 1}$ of $\mu$-distributed random variables and, for all $n \geqslant 1$, the empirical measure*

$$\mu_n := \frac{1}{n}\sum_{k=1}^{n}\delta_{X_k}.$$

*Assume that there exists $\gamma > 0$ such that $\mathcal{E}_{2,\gamma}(\mu) := \int_{\mathbb{R}}\exp\left(\gamma|x|^2\right)\mu(dx) < \infty$. Then for all $n \geqslant 1$, all $x > 0$,*

$$\mathbb{P}\left(\mathcal{W}\left(\mu_n, \mu\right) \geqslant x\right) \leqslant C\exp\left(-cnx^2\right)$$

*where the Wasserstein distance $\mathcal{W}\left(\mu_n, \mu\right)$ is defined by*

$$\mathcal{W}\left(\mu_n, \mu\right) := \inf_{\pi \in \Pi(\mu_n, \mu)}\left\{\int|x - y|\pi(dx, dy)\right\}$$

*and the positive constant $C$ and $c$ depends only on $\gamma$ and $\mathcal{E}_{2,\gamma}(\mu)$.*

**Lemma 6.** *(Lemma 4 (Zhou et al., 2021)). Let $X \sim P$ be a random variable with $X \in [0, M]$, and $P_n$ denotes its empirical distribution of sample size $n$. For $\delta > 0$, for any*

$$\alpha^* \in \arg\max_{\alpha \geqslant 0} \left\{ -\alpha \log \left( \mathbb{E}_P \left[ e^{-X/\alpha} \right] \right) - \alpha\delta \right\} \tag{17}$$

*(1) $\alpha^* = 0$. Furthermore, assume that the support of $X$ is finite. Then there exists a constant $N' := N'(\varepsilon, \delta, P)$, such that $n \geqslant N'$, with probability at least $1 - \varepsilon$, we have*

$$0 \in \arg\max_{\alpha \geqslant 0} \left\{ -\alpha \log \left( \mathbb{E}_{P_n} \left[ e^{-X/\alpha} \right] \right) - \alpha\delta \right\}$$

*(2) $\alpha^* > 0$. Then there exists a constant $N'' := N''(\varepsilon, \delta, P)$, such that for any $n \geqslant N''$, with probability at least $1 - \varepsilon$, there exists a*

$$\widehat{\alpha}^* \in \arg\max_{\alpha \geqslant 0} \left\{ -\alpha \log \left( \mathbb{E}_{P_n} \left[ e^{-X/\alpha} \right] \right) - \alpha\delta \right\}$$

*such that $\alpha^*, \widehat{\alpha}^* \in [\underline{\alpha}, \bar{\alpha}]$, where $\underline{\alpha} > 0$ is independent of $n$ and $\bar{\alpha} = M/\delta$.*

The Total Variation, Chi-square, and Kullback-Liebler uncertainty sets are constructed with the $f$-divergence. The $f$ divergence between the distributions $P$ and $P^o$ is defined as

$$D_f \left( P \| P^o \right) = \int f \left( \frac{dP}{dP^o} \right) dP^o \tag{18}$$

where $f$ is a convex function (Csiszár, 1963). We obtain different divergences for different forms of the function $f$, including some well-known divergences. For example, $f(t) = |t - 1|/2$ gives Total Variation, $f(t) = (t - 1)^2$ gives chi-square, and $f(t) = t \log(t)$ gives Kullback-Liebler.

**Lemma 7.** *(Lemma 5 (Panaganti et al., 2022)) Let $D_f$ be as defined in equation 18 with $f(t) = |t - 1|/2$ corresponding to the TV uncertainty set. Then,*

$$\inf_{D_f(P\|P^o)\leqslant\rho} \mathbb{E}_P[l(X)] = -\inf_{\eta\in\mathbb{R}} \mathbb{E}_{P^o} \left[ (\eta - l(X))_+ \right] + \left( \eta - \inf_{x\in\mathcal{X}} l(x) \right)_+ \times \rho - \eta,$$

**Lemma 8.** *(Covering number (TV)). Given a reward function $R \in \mathcal{R}_{s,a}$, let $\mathcal{U}_R = \{ (\eta \cdot \mathbf{1} - R)_+ : \eta \in [0, R_{\max}] \}$. Fix any $\theta \in (0, 1)$. Denote*

$$\mathcal{N}_R(\theta) = \{ (\eta \cdot \mathbf{1} - R)_+ : \eta \in \{\theta, 2\theta, \dots, N_\theta \cdot \theta\} \}$$

*where $N_\theta = \lceil R_{\max}/\theta \rceil$. Then $\mathcal{N}_R(\theta)$ is a $\theta$-cover for $\mathcal{U}_R$ with respect to $\|\cdot\|_\infty$, and its cardinality is bounded as $|\mathcal{N}_R(\theta)| \leqslant 2R_{\max}/\theta$. Furthermore, for any $\nu \in \mathcal{N}_R(\theta)$, we have $\|\nu\|_\infty \leqslant 1$.*

*Proof.* First, $N_\theta = \lceil R_{\max}/\theta \rceil$ is the minimal number of subintervals of length $\theta$ needed to cover $[0, R_{\max}]$. Denote $J_i = [(i-1)\theta, i\theta)$ to be the $i$-th subinterval, $1 \leqslant i \leqslant N_\theta$. Fix some $\mu \in \mathcal{U}_R$. Then $\mu = (\eta \cdot \mathbf{1} - R)_+$. Without loss of generality, assume this particular $\eta \in J_i$. Let $\nu = ((i\theta) \cdot \mathbf{1} - R)_+$. Now, for any $s, a \in \mathcal{S} \times \mathcal{A}$,

$$\begin{aligned}
|\nu(s, a) - \mu(s, a)| &= |(i\theta - R)_+ - (\eta - R)_+| \\
&\overset{(a)}{\leqslant} |i\theta - R - \eta + R| \\
&\leqslant |i\theta - (i-1)\theta| = \theta
\end{aligned}$$

where (a) follows from $i\theta > \eta$ and the fact that $\max\{x, 0\} - \max\{y, 0\} \leqslant x - y$, if $x > y$. Taking maximum with respect to $s, a$ on both sides, we get $\|\nu - \mu\|_\infty \leqslant \theta$. Since $\nu \in \mathcal{N}_R(\theta)$, this suggests $\mathcal{N}_R(\theta)$ is a $\theta$-cover for $\mathcal{U}_R$. The cardinality bound directly follows from

$$|\mathcal{N}_R(\theta)| = N_\theta = \lceil R_{\max}/\theta \rceil \leqslant R_{\max}/\theta + 1 \leqslant 2R_{\max}/\theta$$

where the last inequality is due to $0 < \theta < 1$. Now, for any $\nu \in \mathcal{N}_R(\theta)$, we can establish the following

$$\nu = (\eta \cdot \mathbf{1} - R)_+ \leqslant (R_{\max}\mathbf{1} - R)_+ \leqslant R_{\max}$$

where the inequality is element-wise.

$\square$

**Lemma 9.** *Fix any* $(s,a) \in \mathcal{S} \times \mathcal{A}$. *Fix any reward function* $R \in \mathcal{R}_{s,a}$. *Let* $\mathcal{N}_R(\theta)$ *be the* $\theta$-*cover of* $\mathcal{U}_R = \{(\eta \cdot \mathbf{1} - R)_+ : \eta \in [0, R_{\max}]\}$ *as described in Lemma 8 . We then have*

$$\sup_{\eta \in [0, R_{\max}]} \left| \mathbb{E}_{R \sim \widehat{\nu}_{s,a}} \left[ (\eta - R)_+ \right] - \mathbb{E}_{R \sim \nu^o_{s,a}} \left[ (\eta - R)_+ \right] \right| \leqslant \max_{r \in \mathcal{N}_R(\theta)} \left| \widehat{\nu}_{s,a} r - \nu^o_{s,a} r \right| + 2\theta$$

*Proof.* For any $\mu \in \mathcal{U}_R$, there exists $r \in \mathcal{N}_R(\theta)$ such that $\|\mu - r\|_\infty \leqslant \theta$. Now for such particular $\mu$ and $r$, we have

$$
\begin{aligned}
\left| \widehat{\nu}_{s,a}\mu - \nu^o_{s,a}\mu \right| &\leqslant \left| \widehat{\nu}_{s,a}\mu - \widehat{\nu}_{s,a}r \right| + \left| \widehat{\nu}_{s,a}r - \nu^o_{s,a}r \right| + \left| \nu^o_{s,a}r - \nu^o_{s,a}\mu \right| \\
&\leqslant \|\widehat{\nu}_{s,a}\|_1 \|\mu - r\|_\infty + \left| \widehat{\nu}_{s,a}r - \nu^o_{s,a}r \right| + \|\nu^o_{s,a}\|_1 \|r - \mu\|_\infty \\
&\leqslant \max_{\nu \in \mathcal{N}_R(\theta)} \left| \widehat{\nu}_{s,a}r - \nu^o_{s,a}r \right| + 2\theta.
\end{aligned}
$$

Taking maximum over $\mathcal{U}_R$ on both sides, we get

$$\sup_{\mu \in \mathcal{U}_R} \left| \widehat{\nu}_{s,a}\mu - \nu^o_{s,a}\mu \right| \leqslant \max_{r \in \mathcal{N}_R(\theta)} \left| \widehat{\nu}_{s,a}r - \nu^o_{s,a}r \right| + 2\theta.$$

Now note that by the definition of $\mathcal{U}_R$, we have

$$\sup_{\eta \in [0, R_{\max}]} \left| \mathbb{E}_{R \sim \widehat{\nu}_{s,a}} \left[ (\eta - R)_+ \right] - \mathbb{E}_{R \sim \nu^o_{s,a}} \left[ (\eta - R)_+ \right] \right| \leqslant \max_{r \in \mathcal{N}_R(\theta)} \left| \widehat{\nu}_{s,a}r - \nu^o_{s,a}r \right| + 2\theta$$

The desired result directly follows.

$\square$

**Lemma 10.** *Consider the* total-variation *uncertainty set*

$$\mathcal{P}^{\mathrm{TV}}_{s,a} = \left\{ P : \tfrac{1}{2} \|P - P^o_{s,a}\|_1 \leqslant \delta \right\}.$$

*Let* $\widehat{R}^{\mathrm{rob}_{TV}}_{s,a} = \min_{r_{s,a} \sim \widehat{\mathcal{R}}_{s,a}} \mathbb{E}_{R \sim r_{s,a}}[R]$ *and* $R^{\mathrm{rob}_{TV}}_{s,a} = \min_{r_{s,a} \sim \mathcal{R}_{s,a}} \mathbb{E}_{R \sim r_{s,a}}[R]$ *be the robust rewards defined using the empirical estimate* $\widehat{\nu}_{s,a}$ *and* $\nu^o_{s,a}$ *and respectively. Then there exists a constant*

$$N^*(\varepsilon, \delta, \nu^o_{s,a}),$$

*such that for all* $n \geq N^*$ *(i.e. a sufficiently large number of reward samples at* $(s,a)$*), the following holds with probability at least* $1 - \varepsilon$:

$$\left| \widehat{R}^{\mathrm{rob}_{TV}}_{s,a} - R^{\mathrm{rob}_{TV}}_{s,a} \right| \leqslant \sqrt{\frac{R^2_{\max} \log(2/\delta)}{2n}}.$$

*Proof.* Following similar analyses as in Proposition 2 (Xu et al., 2023) (Lemma.2), we get

$$\left| \widehat{R}^{\mathrm{rob}_{TV}}_{s,a} - R^{\mathrm{rob}_{TV}}_{s,a} \right| = | \inf_{\eta \in [0, 2R_{\max}/\rho]} \left\{ \mathbb{E}_{R \sim \widehat{\nu}_{s,a}} \left[ (\eta - R)_+ \right] + \left( \eta - \inf_{R' \in [0, R_{\max}]} R' \right)_+ \cdot \rho - \eta \right\} \tag{19}$$

$$- \inf_{\eta \in [0, 2R_{\max}/\rho]} \left\{ \mathbb{E}_{R \sim \nu^o_{s,a}} \left[ (\eta - R)_+ \right] + \left( \eta - \inf_{R' \in [0, R_{\max}]} R' \right)_+ \cdot \rho - \eta \right\} | \tag{20}$$

$$\overset{(a)}{\leqslant} \sup_{\eta \in [0, 2R_{\max}/\rho]} \left| \mathbb{E}_{R \sim \widehat{\nu}_{s,a}} \left[ (\eta - R)_+ \right] - \mathbb{E}_{R \sim \nu^o_{s,a}} \left[ (\eta - R)_+ \right] \right| \tag{21}$$

$$\leqslant \max \left\{ \sup_{\eta \in [0, R_{\max}]} \left| \mathbb{E}_{R \sim \widehat{\nu}_{s,a}} \left[ (\eta - R)_+ \right] - \mathbb{E}_{R \sim \nu^o_{s,a}} \left[ (\eta - R)_+ \right] \right|, \tag{22}$$

$$\sup_{\eta \in [R_{\max}, 2R_{\max}/\rho]} \left| \mathbb{E}_{R \sim \widehat{\nu}_{s,a}} \left[ (\eta - R)_+ \right] - \mathbb{E}_{R \sim \nu^o_{s,a}} \left[ (\eta - R)_+ \right] \right| \right\} \tag{23}$$

$$\overset{(b)}{\leqslant} \max \left\{ \sup_{\eta \in [0, R_{\max}]} \left| \mathbb{E}_{R \sim \widehat{\nu}_{s,a}} \left[ (\eta - R)_+ \right] - \mathbb{E}_{R \sim \nu^o_{s,a}} \left[ (\eta - R)_+ \right] \right|, \tag{24}$$

$$\left| \mathbb{E}_{R \sim \widehat{\nu}_{s,a}} [R] - \mathbb{E}_{R \sim \nu^o_{s,a}} [R] \right| \right\} \tag{25}$$

$$\overset{(c)}{\leqslant} \max \left\{ \max_{r \in \mathcal{N}_R(\theta)} \left| \widehat{\nu}_{s,a} r - \nu^o_{s,a} r \right| + 2\theta, \left| \mathbb{E}_{R \sim \widehat{\nu}_{s,a}} [R] - \mathbb{E}_{R \sim \nu^o_{s,a}} [R] \right| \right\} \tag{26}$$

where $(a)$ follows from the fact that $|\inf_x f(x) - \inf_x g(x)| \leqslant \sup_x |f(x) - g(x)|$. For (b), recall that $R \leqslant R_{\max}$ for any $R \in \mathcal{R}$. Hence, the term $\eta - R'$ is always non-negative for $\eta \in [R_{\max}, 2R_{\max}/\rho]$, which cancels out by linearity of the expectation. (c) follows from applying Lemma 9 to the first term. Recall that all $r \in \mathcal{N}_R(\theta)$ is upper bounded by $R_{\max}$. Now we can apply Hoeffding's inequality to the first term in equation 26:

$$\mathbb{P} \left( \left| \widehat{\nu}_{s,a} r - \nu^o_{s,a} r \right| \geqslant \varepsilon \right) \leqslant 2 \exp \left( -\frac{2n\varepsilon^2}{R^2_{\max}} \right), \quad \forall \varepsilon > 0$$

Now choose $\varepsilon = \sqrt{\frac{R^2_{\max} \log(2|\mathcal{N}_R(\theta)|/\delta)}{2N}}$ and recall that $|\mathcal{N}_R(\theta)| \leqslant 2R_{\max}/\theta$ from Lemma 8. We have

$$\mathbb{P} \left( \left| \widehat{\nu}_{s,a} r - \nu^o_{s,a} r \right| \geqslant \sqrt{\frac{R^2_{\max} \log(4R_{\max}/\theta\delta)}{2n}} \right) \leqslant \mathbb{P} \left( \left| \widehat{\nu}_{s,a} r - \nu^o_{s,a} r \right| \geqslant \sqrt{\frac{R^2_{\max} \log(2 |\mathcal{N}_R(\theta)| /\delta)}{2n}} \right)$$

$$\leqslant \frac{\delta}{|\mathcal{N}_R(\theta)|}$$

Applying a union bound over $\mathcal{N}_R(\theta)$, we get

$$\max_{r \in \mathcal{N}_R(\theta)} \left| \widehat{\nu}_{s,a} r - \nu^o_{s,a} r \right| \leqslant \sqrt{\frac{R^2_{\max} \log(4R_{\max}/\theta\delta)}{2n}} \tag{27}$$

with probability at least $1 - \delta$. Now we can also apply Hoeffding's inequality to the second term in equation 26. Recall that any reward function is bounded by $R_{\max}$. We have

$$\left| \widehat{R}^{\mathrm{rob}_{TV}}_{s,a} - R^{\mathrm{rob}_{TV}}_{s,a} \right| \leqslant \sqrt{\frac{R^2_{\max} \log(2/\delta)}{2n}} \tag{28}$$

with probability at least $1 - \delta$. Combining equation 26 - equation 28 completes the proof.

$\square$

From Lemma 10, we have

$$\mathbb{P}\left(\left|\widehat{R}_{s,a}^{\mathrm{rob}TV} - R_{s,a}^{\mathrm{rob}TV}\right| \geqslant 2\theta + \sqrt{R_{\max}^2 \log(4R_{\max}/\theta\delta)/2n}\right) < \delta \tag{29}$$

Set $\theta = \varepsilon/4$ with $\varepsilon = 2\sqrt{R_{\max}^2 \log(4R_{\max}/\theta\delta)/2n}$, then

$$2n\varepsilon^2/4 = R_{\max}^2 \log(4R_{\max}/\theta\delta) \Rightarrow \exp\{-2n\varepsilon^2/R_{\max}^2\} = \varepsilon\delta/16R_{\max} \tag{30}$$

$$\Rightarrow \delta = \frac{16R_{\max}\exp\{-n\varepsilon^2/2R_{\max}^2\}}{\varepsilon} \tag{31}$$

so that

$$\mathbb{P}\left(\left|\widehat{R}_{s,a}^{\mathrm{rob}TV} - R_{s,a}^{\mathrm{rob}TV}\right| \geqslant \varepsilon\right) < \frac{16R_{\max}\exp\{-n\varepsilon^2/2R_{\max}^2\}}{\varepsilon} \tag{32}$$

**Lemma 11.** *(Lemma 9 (Panaganti et al., 2022)) Let $D_f$ be defined as in equation 18 with the convex function $f(t) = (t-1)^2$ corresponding to the Chi-square uncertainty set. Then*

$$\inf_{D_f(P\|P^\circ)\leqslant\rho} \mathbb{E}_P[l(X)] = -\inf_{\eta\in\mathbb{R}}\left\{\sqrt{\rho+1}\sqrt{\mathbb{E}_{P^\circ}\left[(\eta - l(X))^2\right]} - \eta\right\}$$

**Lemma 12.** *Fix any $s, a \in \mathcal{S} \times \mathcal{A}$. For any $\theta, \delta \in (0,1)$ and $\rho > 0$, we have, with probability at least $1 - \delta$, we can find a constant $N^\star$ such that $\forall n \geqslant N^\star$, we have*

$$\left|\widehat{R}_{s,a}^{\mathrm{rob}\chi} - R_{s,a}^{\mathrm{rob}\chi}\right| \leqslant 2\theta + \frac{\sqrt{2}C_\rho^2 R_{\max}}{(C_\rho - 1)\sqrt{n}}\left(\sqrt{\log\left(\frac{2\left(1 + C_\rho R_{\max}/\left(\theta\left(C_\rho - 1\right)\right)\right)}{\delta}\right)} + 1\right).$$

*Proof.* Similar to Lemma 10, the result is direct by applying the results of Lemma 9, Lemma 11 and the law of total probability. $\square$

From the results of Lemma 12, Then we have

$$\mathbb{P}\left(\left|\widehat{R}_{s,a}^{\mathrm{rob}\chi} - R_{s,a}^{\mathrm{rob}\chi}\right| \geqslant 2\theta + \frac{\sqrt{2}C_p^2 R_{\max}}{(C_p - 1)\sqrt{n}}\left(\sqrt{\log\left(\frac{2(1 + C_p R_{\max}/(\theta(C_p - 1)))}{\delta}\right)} + 1\right)\right) < \delta \tag{33}$$

Set $\theta = \varepsilon/4$ with $\varepsilon = 2\frac{\sqrt{2}C_p^2 R_{\max}}{(C_p - 1)\sqrt{n}}\left(\sqrt{\log\left(\frac{2(1 + C_p R_{\max}/(\theta(C_p - 1)))}{\delta}\right)} + 1\right)$, then

$$\left(\frac{(C_p - 1)\sqrt{n}\varepsilon}{2\sqrt{2}C_p^2 R_{\max}} - 1\right)^2 = \log\left(\frac{2(1 + C_p R_{\max}/(\theta(C_p - 1)))}{\delta}\right) \tag{34}$$

$$\Rightarrow \exp\left\{-\left(\frac{(C_p - 1)\sqrt{n}\varepsilon}{2\sqrt{2}C_p^2 R_{\max}} - 1\right)^2\right\} = \frac{\delta}{2(1 + C_p R_{\max}/(\theta(C_p - 1)))} \tag{35}$$

$$\Rightarrow \mathbb{P}\left(\left|\widehat{R}_{s,a}^{\mathrm{rob}\chi} - R_{s,a}^{\mathrm{rob}\chi}\right| \geqslant \varepsilon\right) < 2\left(1 + C_p R_{\max}/(\frac{\varepsilon(C_p - 1)}{4})\right)\exp\left\{-\left(\frac{(C_p - 1)\sqrt{n}\varepsilon}{2\sqrt{2}C_p^2 R_{\max}} - 1\right)^2\right\} \tag{36}$$

**Lemma 13.** *Consider an MDP with the Wasserstein distance $D_W$. Fix any $s, a \in \mathcal{S} \times \mathcal{A}$, we can derive*

$$\inf_{D_W(P\|P^\circ)\leqslant\rho} \mathbb{E}_P[l(X)] = -\inf_{\lambda\in[0,R_{\max}/\rho^p]}\left(\lambda\rho^p - \mathbb{E}_{R'\sim\nu_{s,a}^\circ}\left[\inf_{R''\in\mathcal{R}}\left\{R'' + \lambda d^p\left(R'', R'\right)\right\}\right]\right).$$

*Proof.* Fix any $(s, a) \in \mathcal{S} \times \mathcal{A}$. We have

$$
\inf_{D_W(P \| P^o) \leqslant \rho} \mathbb{E}_{R \sim P}[l(X)] = - \sup_{D_W(P \| P^o) \leqslant \rho} \mathbb{E}_{R \sim P}[-l(X)]
$$

$$
\overset{(a)}{=} - \inf_{\lambda \geqslant 0} \left\{ \mathbb{E}_{R' \sim \nu_{s,a}^o} \left[ \sup_{R'' \in \mathcal{R}} \{ -R'' - \lambda d^p(R'', R') \} \right] + \lambda \rho^p \right\}
$$

$$
= \sup_{\lambda \geqslant 0} \left\{ \mathbb{E}_{R' \sim \nu_{s,a}^o} \left[ \inf_{R'' \in \mathcal{R}} \{ R'' + \lambda d^p(R'', R') \} \right] - \lambda \rho^p \right\}
$$

$$
\overset{(b)}{=} \sup_{\lambda \in [0, R_{\max}/\rho^p]} \left\{ \mathbb{E}_{R' \sim \nu_{s,a}^o} \left[ \inf_{R'' \in \mathcal{R}} \{ R'' + \lambda d^p(R'', R') \} \right] - \lambda \rho^p \right\},
$$

where $(a)$ follows from ((Gao and Kleywegt, 2023) Theorem 1). For $(b)$, let us first denote any optimizer in $(a)$ to be $\lambda^*$. Observe that since $R$ is non-negative, it follows that

$$
0 \leqslant -\lambda^* \rho^p + \mathbb{E}_{R' \sim \nu_{s,a}^o} \left[ \inf_{R''} \{ R'' + \lambda d^p(R'', R') \} \right] \leqslant -\lambda^* \rho^p + \mathbb{E}_{R' \sim \nu_{s,a}^o} [R' + \lambda d^p(R', R')] \leqslant -\lambda^* \rho^p + R_{\max}
$$

where in the last inequality we use that the distance metric satisfies $d(R, R) = 0$, for any $R \in \mathcal{R}$.

$\square$

**Lemma 14.** *(Covering number (Wasserstein)). Consider the following set of $\mathbb{R}^{|\mathcal{R}|}$ vectors:*

$$
\mathcal{U}_{\rho, R} = \left\{ \left( \inf_{R'' \in \mathcal{R}} \{ R'' + \lambda d^p(R'', 1) \}, \ldots, \inf_{R'' \in \mathcal{R}} \{ R'' + \lambda d^p(R'', |\mathcal{R}|) \} \right)^T : \lambda \in [0, R_{\max}/\rho^p] \right\}
$$

*Let*

$$
\mathcal{N}_{\rho, R}(\theta) = \left\{ \left( \inf_{R'' \in \mathcal{R}} \{ R'' + \lambda d^p(R'', 1) \}, \ldots, \inf_{R'' \in \mathcal{R}} \{ R'' + \lambda d^p(R'', |\mathcal{R}|) \} \right)^T : \lambda \in \left\{ \frac{\theta}{B_p}, \frac{2\theta}{B_p}, \ldots, N_{\rho, \theta} \frac{\theta}{B_p} \right\} \right\},
$$

*where $N_{\rho, \theta} = \left\lceil \frac{R_{\max} B_p}{\rho^p \theta} \right\rceil$ and $B_p = \max_{R', R''} d^p(R'', R')$. Then $\mathcal{N}_{\rho, R}(\theta)$ is a $\theta$-cover of $\mathcal{U}_{\rho, R}$ with respect to $\| \cdot \|_\infty$, and its cardinality is bounded as $|\mathcal{N}_{\rho, R}(\theta)| \leqslant \frac{R_{\max} B_p + (R_{\max} \vee \rho^p)}{\rho^p \theta}$. Furthermore, for any $\nu \in \mathcal{N}_{\rho, R}(\theta)$, we have $\| \nu \|_\infty \leqslant \frac{R_{\max}(B_p + \rho^p)}{\rho^p}$.*

*Proof.* Fix any $\theta \in (0, 1)$. First note that $N_{\rho, \theta}$ is the minimal number of subintervals of length $\frac{\theta}{B_p}$ needed to cover $[0, R_{\max}/\rho^p]$. Denote $J_i = \left[ (i-1) \frac{\theta}{B_p}, i \frac{\theta}{B_p} \right), 1 \leqslant i \leqslant N_{\rho, \theta}$. Fix some $\mu \in \mathcal{U}_{\rho, R}$. Then $\mu$ must takes the form

$$
\mu = \left( \inf_{R'' \in \mathcal{R}} \{ R'' + \lambda d^p(R'', 1) \}, \ldots, \inf_{R'' \in \mathcal{R}} \{ R'' + \lambda d^p(R'', |\mathcal{R}|) \} \right)^T,
$$

for some $\lambda \in [0, R_{\max}/\rho^p]$. Without loss of generality, assume $\lambda \in J_i$. Now we pick

$$
\nu = \left( \inf_{R'' \in \mathcal{R}} \left\{ R'' + i \frac{\theta}{B_p} d^p(R'', 1) \right\}, \ldots, \inf_{R'' \in \mathcal{R}} \left\{ R'' + i \frac{\theta}{B_p} d^p(R'', |\mathcal{R}|) \right\} \right)^T
$$

Fix any $R' \in \mu$ and $R'' \in \nu$, we have

$$\begin{aligned}
|R' - R''| &= \left| \inf_{R'' \in \mathcal{R}} \{R'' + \lambda d^p (R'', R')\} - \inf_{R'' \in \mathcal{R}} \left\{ R'' + i \frac{\theta}{B_p} d^p (R'', R) \right\} \right| \\
&\overset{(a)}{\leqslant} \sup_{R'' \in \mathcal{R}} \left| \left( \lambda - i \frac{\theta}{B_p} \right) d^p (R'', R') \right| \\
&\leqslant \left| \lambda - i \frac{\theta}{B_p} \right| \max_{R', R''} d^p (R'', R') = \left| \lambda - i \frac{\theta}{B_p} \right| B_p \\
&\leqslant \left| (i - 1) \frac{\theta}{B_p} - i \frac{\theta}{B_p} \right| B_p = \theta
\end{aligned}$$

where $(a)$ is due to $|\inf_x f(x) - \inf_x g(x)| \leqslant \sup_x |f(x) - g(x)|$. Taking maximum over $R' \in \mathcal{R}$ on both sides, we get $\|\mu - \nu\|_\infty \leqslant \theta$. Since $\nu \in \mathcal{N}_{\rho,R}(\theta)$, this suggests that $\mathcal{N}_{\rho,R}(\theta)$ is a $\theta$-cover for $\mathcal{U}_{\rho,R}$.

To bound the cardinality of $\mathcal{N}_{\rho,R}(\theta)$, we consider two cases. If $0 < \rho < 1$, then $\rho^p \theta < 1$ and

$$\left\lceil \frac{R_{\max} B_p}{\rho^p \theta} \right\rceil \leqslant \frac{R_{\max} B_p}{\rho^p \theta} + 1 \leqslant \frac{R_{\max} B_p}{\rho^p \theta} + \frac{R_{\max}}{\rho^p \theta} = \frac{R_{\max} B_p + R_{\max}}{\rho^p \theta}$$

On the other hand, if $\rho > 1$, then since $\theta \in (0, 1)$, we have

$$\left\lceil \frac{R_{\max} B_p}{\rho^p \theta} \right\rceil \leqslant \frac{R_{\max} B_p}{\rho^p \theta} + 1 = \frac{R_{\max} B_p}{\rho^p \theta} + \frac{\rho^p \theta}{\rho^p \theta} \leqslant \frac{R_{\max} B_p}{\rho^p \theta} + \frac{\rho^p}{\rho^p \theta} = \frac{R_{\max} B_p + \rho^p}{\rho^p \theta}$$

Hence, we have $|\mathcal{N}_{\rho,R}(\theta)| = N_{\rho,\theta} \leqslant \frac{R_{\max} B_p + (R_{\max} \vee \rho^p)}{\rho^p \theta}$. Now we prove the last claim. Fix any $\nu \in \mathcal{N}_{\rho,R}$. Note that for any $R' \in \mathcal{R}$,

$$R' = \inf_{R'' \in \mathcal{R}} \{R'' + \lambda d^p (R'', R')\} \leqslant R_{\max} + \lambda B_p \leqslant R_{\max} + \frac{R_{\max}}{\rho^p} B_p = \frac{R_{\max} (B_p + \rho^p)}{\rho^p}$$

The result then follows from taking maximum over $R' \in \mathcal{R}$ on both sides. $\qquad \square$

**Lemma 15.** *Fix any $(s, a) \in \mathcal{S} \times \mathcal{A}$. Let $\mathcal{N}_{\rho,R}(\theta)$ be the $\theta$-cover of the set*

$$\mathcal{U}_{\rho,R} = \left\{ \left( \inf_{R'' \in \mathcal{R}} \{R'' + \lambda d^p (R'', 1)\}, \dots, \{R'' + \lambda d^p (R'', |\mathcal{R}|)\} \right)^T : \lambda \in [0, R_{\max}/\rho^p] \right\}$$

*as described in Lemma 14. We then have*

$$\sup_{\lambda \in [0, R_{\max}/\rho^p]} | \mathbb{E}_{R' \sim \nu^o_{s,a}} \left[ \inf_{R'' \in \mathcal{R}} \{R'' + \lambda d^p (R'', R')\} \right] - \mathbb{E}_{R' \sim \widehat{\nu}^o_{s,a}} \left[ \inf_{R'' \in \mathcal{R}} \{R'' + \lambda d^p (R'', R')\} \right] |$$
$$\leqslant \max_{r \in \mathcal{N}_{\rho,R}(\theta)} \left| \widehat{\nu}^o_{s,a} r - \nu^o_{s,a} r \right| + 2\theta.$$

*Proof.* The proof is identical to the proof of Lemma 9. $\qquad \square$

**Lemma 16.** *Fix any $(s, a) \in \mathcal{S} \times \mathcal{A}$. For any $\theta, \delta \in (0, 1)$ and $\rho > 0$, we have the following inequality with probability at least $1 - \delta$*

$$\left| \widehat{R}^{\mathrm{rob}\mathcal{W}}_{s,a} - R^{\mathrm{rob}\mathcal{W}}_{s,a} \right| \leqslant \frac{R_{\max} (B_p + \rho^p)}{\rho^p} \sqrt{\frac{\log \left( \frac{2 R_{\max} B_p + 2 (R_{\max} \vee \rho^p)}{\rho^p \theta \delta} \right)}{2n}} + 2\theta$$

*where $B_p = \max_{R', R''} d^p (R'', R')$.*

*Proof.* From Lemma 13 , we have

$$R^{\text{rob}\mathcal{W}}_{s,a} = \sup_{\lambda \in [0, R_{\max}/\rho^p]} \left\{ \mathbb{E}_{R' \sim \nu^o_{s,a}} \left[ \inf_{R'' \in \mathcal{R}} \{ R'' + \lambda d^p (R'', R') \} \right] - \lambda \rho^p \right\},$$

$$\widehat{R}^{\text{rob}\mathcal{W}}_{s,a} = \sup_{\lambda \in [0, R_{\max}/\rho^p]} \left\{ \mathbb{E}_{R' \sim \widehat{\nu}^o_{s,a}} \left[ \inf_{R'' \in \mathcal{R}} \{ R'' + \lambda d^p (R'', R') \} \right] - \lambda \rho^p \right\}.$$

Now it follows that

$$\left| \widehat{R}^{\text{rob}\mathcal{W}}_{s,a} - R^{\text{rob}\mathcal{W}}_{s,a} \right| = | \sup_{\lambda \in [0, R_{\max}/\rho^p]} \left\{ \mathbb{E}_{R' \sim \nu^o_{s,a}} \left[ \inf_{R'' \in \mathcal{R}} \{ R'' + \lambda d^p (R'', R') \} \right] - \lambda \rho^p \right\} \tag{37}$$

$$- \sup_{\lambda \in [0, R_{\max}/\rho^p]} \left\{ \mathbb{E}_{R' \sim \widehat{\nu}^o_{s,a}} \left[ \inf_{R'' \in \mathcal{R}} \{ R'' + \lambda d^p (R'', R') \} \right] - \lambda \rho^p \right\} | \tag{38}$$

$$\overset{(a)}{\leqslant} \sup_{\lambda \in [0, R_{\max}/\rho^p]} | \mathbb{E}_{R' \sim \nu^o_{s,a}} \left[ \inf_{R'' \in \mathcal{R}} \{ R'' + \lambda d^p (R'', R') \} \right] \tag{39}$$

$$- \mathbb{E}_{R' \sim \widehat{\nu}^o_{s,a}} \left[ \inf_{R'' \in \mathcal{R}} \{ R'' + \lambda d^p (R'', R') \} \right] | \tag{40}$$

$$\overset{(b)}{\leqslant} \max_{r \in \mathcal{N}_{\rho, R}(\theta)} \left| \widehat{\nu}^o_{s,a} r - \nu^o_{s,a} r \right| + 2\theta \tag{41}$$

where $(a)$ follows from $|\sup_x f(x) - \sup_x g(x)| \leqslant \sup_x |f(x) - g(x)|$. (b) follows from Lemma 15.
Recall that all $\nu \in \mathcal{N}_{\rho, R}(\theta)$ is bounded by $\nu_{\max} := \frac{R_{\max}(B_p + \rho^p)}{\rho^p}$. Now we can apply Hoeffding's inequality:

$$\mathbb{P}\left( \left| \widehat{\nu}^o_{s,a} r - \nu^o_{s,a} r \right| \geqslant \varepsilon \right) \leqslant 2 \exp\left( -\frac{2n\varepsilon^2}{\nu^2_{\max}} \right) = 2 \exp\left( -\frac{2n\varepsilon^2}{\left( \frac{R_{\max}(B_p + \rho^p)}{\rho^p} \right)^2} \right), \quad \forall \varepsilon > 0$$

Now recall that $|\mathcal{N}_{\rho, R}(\theta)| \leqslant \frac{R_{\max} B_p + (R_{\max} \vee \rho^p)}{\rho^p \theta}$ and choose

$$\varepsilon = \frac{R_{\max}(B_p + \rho^p)}{\rho^p} \sqrt{\frac{\log(2 |\mathcal{N}_{\rho, R}(\theta)| /\delta)}{2N}}$$

We then have

$$\mathbb{P}\left( \left| \widehat{\nu}^o_{s,a} r - \nu^o_{s,a} r \right| \geqslant \frac{R_{\max}(B_p + \rho^p)}{\rho^p} \sqrt{\frac{\log(2 |\mathcal{N}_{\rho, R}(\theta)| /\delta)}{2n}} \right) \leqslant \frac{\delta}{|\mathcal{N}_{\rho, R}(\theta)|}.$$

Finally, applying a union bound over $\mathcal{N}_{\rho, R}(\theta)$, we get

$$\max_{r \in \mathcal{N}_{\rho, R}(\theta)} \left| \widehat{\nu}^o_{s,a} r - \nu^o_{s,a} r \right| \leqslant \frac{R_{\max}(B_p + \rho^p)}{\rho^p} \sqrt{\frac{\log\left( \frac{2R_{\max} B_p + 2(R_{\max} \vee \rho^p)}{\rho^p \theta \delta} \right)}{2n}},$$

with probability at least $1 - \delta$. Combining the above and equation 41 completes the proof. $\square$

Similarly, Set $\theta = \varepsilon/4$ with $\varepsilon = \frac{2R_{\max}(B_p + \rho^p)}{\rho^p} \sqrt{\frac{\log\left(\frac{2R_{\max}B_p + 2R_{\max}\sqrt{\rho^p}}{\rho^p \theta \delta}\right)}{2n}}$, then

$$\exp\left\{-\frac{2n\varepsilon^2 \rho^{2p}}{4R_{\max}^2 (B_p + \rho^p)^2}\right\} = \frac{\rho^p \theta \delta}{2R_{\max}B_p + 2R_{\max}\sqrt{\rho^p}} \tag{42}$$

$$\Rightarrow \delta = \frac{4\left(2R_{\max}B_p + 2R_{\max}\sqrt{\rho^p}\right)}{\rho^p \varepsilon} \exp\left\{-\frac{2n\varepsilon^2 \rho^{2p}}{4R_{\max}^2(B_p + \rho^p)^2}\right\} \tag{43}$$

so that

$$\mathbb{P}\left(\left|\widehat{R}_{s,a}^{\text{rob}\mathcal{W}} - R_{s,a}^{\text{rob}\mathcal{W}}\right| \geqslant \varepsilon\right) < \frac{4\left(2R_{\max}B_p + 2R_{\max}\sqrt{\rho^p}\right)}{\rho^p \varepsilon} \exp\left\{-\frac{2n\varepsilon^2 \rho^{2p}}{4R_{\max}^2(B_p + \rho^p)^2}\right\}. \tag{44}$$

## C. Convergence of `Robust-Power-UCT` Multi-armed bandits

**Lemma 17.** *For $m \in [M]$, let $(\widehat{V}_{m,n})_{n \geqslant 1}$ be a sequence of estimator satisfying $\widehat{V}_{m,n} \overset{\alpha,\beta}{\underset{n\to\infty}{\to}} V_m$, and there exists a constant $L$ such that $\widehat{V}_{m,n} \leqslant L, \forall n \geqslant 1$. Let $X_i$ be an iid sequence from a distribution $\nu^o$ with mean $\mu$ and $S_i$ be an iid sequence from a distribution $p = (p_1, \ldots, p_M)$ supported on $\{1, \ldots, M\}$. Introducing the random variables $N_m^n = \#|\{i \leqslant n : S_i = s_m\}|$. Let us study a random vector $\widehat{p}_n = (\frac{N_1^n}{n}, \frac{N_2^n}{n}, \ldots, \frac{N_M^n}{n})$. We define an estimate of $\nu^o$ as*

$$\nu^o \approx \widehat{\nu}_n = \frac{1}{n}\sum_{i=1}^n \delta_{X_i},$$

*where $\delta_{X_i}$ is a point mass at $X_i$. And define $R^{\mathcal{R}} = \min_{r \sim \mathcal{R}} \mathbb{E}_{R \sim r}[R]$ w.r.t $\nu^o$ and $R^{\widehat{\mathcal{R}}} = \min_{r \sim \widehat{\mathcal{R}}} \mathbb{E}_{R \sim r}[R]$ w.r.t $\widehat{\nu}_n$. We define the sequence of estimator*

$$\widehat{Q}_n = R^{\widehat{\mathcal{R}}} + \gamma \sigma_{\widehat{p}_n}(\widehat{V}_n).$$

*Then with $2\alpha \leqslant \beta, \beta > 1$,*

$$\widehat{Q}_n \overset{\alpha,\beta}{\underset{n\to\infty}{\to}} R^{\mathcal{R}} + \gamma \sigma_p(V).$$

*Proof.* Let $p = (p_1, p_2, \ldots p_M), p \in \triangle^M$ where $\triangle^M = \{x \in \mathbb{R}^M : \sum_{i=1}^M x_i = 1, x_i \geqslant 0\}$ is the $(M-1)$-dimensional simplex. Without loss of generality, we assume that $p_m > 0$ for all $m$. Let us define $V = (V_1, V_2, \ldots V_M)$. Let $\widehat{V}_n = (\widehat{V}_{1,N_1^n}, \widehat{V}_{2,N_2^n}, \ldots, \widehat{V}_{M,N_M^n}), \sum_{i=1}^M N_i^n = n, N_i^n$ is the number of times that population $i$ was observed. We have $\widehat{Q}_n = R^{\widehat{\mathcal{R}}} + \gamma \sigma_{\widehat{p}_n}(\widehat{V}_n)$. Therefore,

$$\mathbb{P}\left(|\widehat{Q}_n - (R^{\mathcal{R}} + \gamma \sigma_p(V))| \geqslant \varepsilon\right) \leqslant \mathbb{P}\left(|R^{\widehat{\mathcal{R}}} - R^{\mathcal{R}}| \geqslant \frac{1}{2}\varepsilon\right) + \mathbb{P}\left(|\gamma \sigma_{\widehat{p}_n}(\widehat{V}_n) - \gamma \sigma_p(V)| \geqslant \frac{1}{2}\varepsilon\right) \tag{45}$$

$$\leqslant \underbrace{\mathbb{P}\left(|R^{\widehat{\mathcal{R}}} - R^{\mathcal{R}}| \geqslant \frac{1}{2}\varepsilon\right)}_{A} + \underbrace{\mathbb{P}\left(|\sigma_{\widehat{p}_n}(\widehat{V}_n) - \sigma_p(V)| \geqslant \frac{1}{2\gamma}\varepsilon\right)}_{B}. \tag{46}$$

To upper bound A,

- TV:

$$A \leqslant \frac{16R_{\max}\exp\{-n(\varepsilon/4)^2/2R_{\max}^2\}}{\varepsilon/4} \tag{47}$$

- Chi-square:

$$A \leqslant 2\left(1 + C_p R_{\max}/(\frac{\varepsilon(C_p - 1)}{4})\right)\exp\left\{-\left(\frac{(C_p - 1)\sqrt{n}\varepsilon}{2\sqrt{2}C_p^2 R_{\max}} - 1\right)^2\right\} \tag{48}$$

- Wasserstein:

$$A \leqslant \frac{4 \left(2R_{\max} B_p + 2R_{\max}\sqrt{\rho^p}\right)}{\rho^p \varepsilon} \exp\left\{ -\frac{2n\varepsilon^2 \rho^{2p}}{4R_{\max}^2 (B_p + \rho^p)^2} \right\} \tag{49}$$

To upper bound B, let us consider $\sigma_{\widehat{p}_n}(\widehat{V}_n) - \sigma_p(V) = (\sigma_{\widehat{p}_n}(\widehat{V}_n) - \sigma_p(\widehat{V}_n)) + (\sigma_p(\widehat{V}_n) - \sigma_p(V))$. Then,

$$B \leqslant \underbrace{\mathbb{P}\left( \left|\sigma_{\widehat{p}_n}(\widehat{V}_n) - \sigma_p(\widehat{V}_n)\right| \geqslant \frac{1}{4\gamma}\varepsilon \right)}_{B_1} + \underbrace{\mathbb{P}\left( \left|\sigma_p(\widehat{V}_n) - \sigma_p(V)\right| \geqslant \frac{1}{4\gamma}\varepsilon \right)}_{B_2}. \tag{50}$$

By applying results from Lemma 2 and Equation 8, we obtain

- TV:

$$B_1 \leqslant \frac{16H \exp\{-n(\varepsilon/4\gamma)^2/2H^2\}}{\varepsilon/4\gamma} \tag{51}$$

- Chi-square:

$$B_1 \leqslant 2 \left( 1 + C_p H / (\frac{\varepsilon(C_p - 1)}{4}) \right) \exp\left\{ -\left( \frac{(C_p - 1)\sqrt{n}\varepsilon}{2\sqrt{2}C_p^2 H} - 1 \right)^2 \right\} \tag{52}$$

- Wasserstein:

$$B_1 \leqslant \frac{4 \left(2HB_p + 2H\sqrt{\rho^p}\right)}{\rho^p \varepsilon} \exp\left\{ -\frac{2n\varepsilon^2 \rho^{2p}}{4H^2 (B_p + \rho^p)^2} \right\} \tag{53}$$

For $B_2$, as the result from Lemma 1, we have

$$\left|\sigma_p(\widehat{V}_n) - \sigma_p(V)\right| \leqslant \|\widehat{V}_n - V\|_\infty \leqslant \|\widehat{V}_n - V\|_1$$

Therefore,

$$B_2 \leqslant \mathbf{Pr}\left( \sum_{m=1}^{M} |\widehat{V}_{m,N_m^n} - V_m| \geqslant \frac{1}{4\gamma}\varepsilon \right) \leqslant \sum_{m=1}^{M} \mathbf{Pr}\left( |\widehat{V}_{m,N_m^n} - V_m| \geqslant \frac{1}{4\gamma M}\varepsilon \big| N_m^n \right) \tag{54}$$

$$\leqslant \sum_{m=1}^{M} \mathbb{E}\left[ \mathbb{P}\left( \frac{1}{N_m^n} \sum_{t=1}^{N_m^n} V_{m,t} - V_m \geqslant \frac{1}{4\gamma M}\varepsilon \big| N_m^n \right) \right] \tag{55}$$

$$\leqslant \sum_{m=1}^{M} \mathbb{E}\left[ c(N_m^n)^{-\alpha} (\frac{\varepsilon}{4\gamma M})^{-\beta} \right]. \tag{56}$$

Let us define an event $\mathcal{E} = \left\{ N_m^n > \frac{np_m}{2} \right\}$. Therefore,

$$B_2 \leqslant \sum_{m=1}^{M} \mathbb{E}\left[ c(\frac{np_m}{2})^{-\alpha} (\frac{\varepsilon}{4\gamma M})^{-\beta} \right] + \sum_{m=1}^{M} \mathbb{E}\left[ \mathbb{P}(N_m^n \leqslant \frac{np_m}{2}) \right] \tag{57}$$

$$= \sum_{m=1}^{M} (c2^{\alpha+2\beta}\gamma^\beta p_m^{-\alpha} M^\beta)n^{-\alpha}\varepsilon^{-\beta} + \sum_{m=1}^{M} \mathbb{E}\left[ \mathbb{P}(N_m^n - p_m n \leqslant -\frac{p_m n}{2}) \right] \tag{58}$$

$$\leqslant \sum_{m=1}^{M} (c2^{\alpha+2\beta}\gamma^\beta p_m^{-\alpha} M^\beta)n^{-\alpha}\varepsilon^{-\beta} + \sum_{m=1}^{M} \exp\left\{ -2n(\frac{p_m n}{2})^2 \right\} \tag{59}$$

Therefore,

- TV:

$$A + B \leqslant A + B_1 + B_2 \leqslant \frac{16R_{\max}\exp\{-n(\varepsilon/4)^2/2R_{\max}^2\}}{\varepsilon/4} + \frac{16H\exp\{-n(\varepsilon/4\gamma)^2/2H^2\}}{\varepsilon/4\gamma} \tag{60}$$

$$+ \sum_{m=1}^{M}(c2^{\alpha+2\beta}\gamma^\beta p_m^{-\alpha}M^\beta)n^{-\alpha}\varepsilon^{-\beta} + \sum_{m=1}^{M}\exp\left\{-2n(\frac{p_m n}{2})^2\right\}. \tag{61}$$

- Chi-Square:

$$A + B \leqslant A + B_1 + B_2 \leqslant 2\left(1 + C_p R_{\max}/(\frac{\varepsilon(C_p-1)}{4})\right)\exp\left\{-\left(\frac{(C_p-1)\sqrt{n}\varepsilon}{2\sqrt{2}C_p^2 R_{\max}}-1\right)^2\right\} \tag{62}$$

$$+ 2\left(1 + C_p H/(\frac{\varepsilon(C_p-1)}{4})\right)\exp\left\{-\left(\frac{(C_p-1)\sqrt{n}\varepsilon}{2\sqrt{2}C_p^2 H}-1\right)^2\right\} + \sum_{m=1}^{M}(c2^{\alpha+2\beta}\gamma^\beta p_m^{-\alpha}M^\beta)n^{-\alpha}\varepsilon^{-\beta} \tag{63}$$

$$+ \sum_{m=1}^{M}\exp\left\{-2n(\frac{p_m n}{2})^2\right\}. \tag{64}$$

- Wasserstein:

$$A + B \leqslant A + B_1 + B_2 \leqslant \frac{4\left(2R_{\max}B_p + 2R_{\max}\sqrt{\rho^p}\right)}{\rho^p\varepsilon}\exp\left\{-\frac{2n\varepsilon^2\rho^{2p}}{4R_{\max}^2(B_p+\rho^p)^2}\right\} \tag{65}$$

$$+ \frac{4\left(2HB_p + 2H\sqrt{\rho^p}\right)}{\rho^p\varepsilon}\exp\left\{-\frac{2n\varepsilon^2\rho^{2p}}{4H^2(B_p+\rho^p)^2}\right\} + \sum_{m=1}^{M}(c2^{\alpha+2\beta}\gamma^\beta p_m^{-\alpha}M^\beta)n^{-\alpha}\varepsilon^{-\beta} \tag{66}$$

$$+ \sum_{m=1}^{M}\exp\left\{-2n(\frac{p_m n}{2})^2\right\}. \tag{67}$$

In both three cases, that leads to

$$\mathbb{P}\left(|\widehat{Q}_n - (R^{\mathcal{R}} + \gamma\sigma_p(V))| \geqslant \varepsilon\right) \leqslant \mathcal{O}(\exp\{-n\}) + \mathcal{O}(e^{-1}\exp\{-cn\varepsilon^2\}) + \sum_{m=1}^{M}(c2^{\alpha+2\beta}\gamma^\beta p_m^{-\alpha}M^\beta)n^{-\alpha}\varepsilon^{-\beta} \tag{68}$$

$$+ \sum_{m=1}^{M}\exp\left\{-2n(\frac{p_m n}{2})^2\right\} \leqslant c^{'}n^{-\alpha}\varepsilon^{-\beta}, \tag{69}$$

with $c^{'} > 0$ depends on $c, M, \alpha, \beta, p_i$. Here we need

$$2\alpha \leqslant \beta, \tag{70}$$

to argue that $e^{-1}\exp(-cn\varepsilon^2) = \mathcal{O}(n^{-\alpha}\varepsilon^{-\beta})$. Therefore, with $n \geqslant 1, \varepsilon > 0$,

$$\mathbb{P}\left(\left|\widehat{Q}_n - (R^{\mathcal{R}} + \gamma\sigma_p(V))\right| \geqslant \varepsilon\right) \leqslant c^{'}n^{-\alpha}\varepsilon^{-\beta}. \tag{71}$$

Furthermore,

$$\lim_{n \longrightarrow \infty} \mathbb{E}\left[\left|\widehat{Q}_n - \left(R^{\mathcal{R}} + \gamma \sigma_p(V)\right)\right|\right] \tag{72}$$

$$= \lim_{n \longrightarrow \infty} \int_0^\infty \mathbb{P}\left(\left|\widehat{Q}_n - \left(R^{\mathcal{R}} + \gamma \sigma_p(V)\right)\right| \geqslant s\right) ds \tag{73}$$

$$\leqslant \lim_{n \longrightarrow \infty} \left(\int_0^{n^{-\frac{\alpha}{\beta}}} 1 ds + \int_{n^{-\frac{\alpha}{\beta}}}^{+\infty} c' n^{-\alpha} s^{-\beta}\right) \tag{74}$$

$$= \lim_{n \longrightarrow \infty} \left(n^{-\frac{\alpha}{\beta}} + c' n^{-\alpha}\left(\frac{s^{-\beta+1}}{-\beta+1} + C\right)\Big|_{n^{-\frac{\alpha}{\beta}}}^{+\infty}\right) = 0 \tag{75}$$

$$= \lim_{n \longrightarrow \infty} \left(n^{-\frac{\alpha}{\beta}} - c' n^{-\alpha}\left(\frac{n^{\frac{\alpha(\beta-1)}{\beta}}}{-\beta+1}\right)\right) = 0 \text{ (because } \alpha > 0, \beta > 1) \tag{76}$$

so that,

$$\lim_{n \to \infty} \mathbb{E}[\widehat{Q}_n] = R^{\mathcal{R}} + \gamma \sigma_p(V).$$

This means

$$\widehat{Q}_n \xrightarrow[n \to \infty]{\alpha,\beta} R^{\mathcal{R}} + \gamma \sigma_p(V),$$

which concludes the proof. □

## D. Convergence of `Robust-Power-UCT` n Monte-Carlo Tree Search

**Theorem 1.** *(Theorem 1 of Dam et al. (2024b)) For each arm $a \in [K]$, let $\widehat{\mu}_{a,n} \xrightarrow[n \to \infty]{\alpha,\beta} \mu_a$ and define $\mu_\star = \max_a\{\mu_a\}$. Suppose arms are selected according to equation 4 with parameters $(\alpha, \beta, b, C)$, and let $p \in [1, \infty)$. If*

$$1 \leq p \leq 2, \text{ and } \alpha \leq \tfrac{\beta}{2}, \quad or \quad p > 2, \ 0 < \alpha - \tfrac{\beta}{p} < 1,$$

*and*

$$\alpha\left(1 - \tfrac{b}{\alpha}\right) \leq b < \alpha,$$

*then there exists a suitable constant $C$ (depending on $K, b, \alpha, p, \Delta_{\min}$) such that*

$$\widehat{\mu}_n(p) \xrightarrow[n \to \infty]{\alpha',\beta'} \mu_\star,$$

*where $\Delta_{\min} = \min_{a: \mu_a < \mu_\star}(\mu_\star - \mu_a)$, $\alpha' = (b-1)\left(1 - \tfrac{b}{\alpha}\right)$, and $\beta' = (b-1)$.*

**Theorem 2.** *When applying `Robust-Power-UCT` with parameters $\{b_i\}_{i=0}^H$, $\{\alpha_i\}_{i=0}^H$, and $\{\beta_i\}_{i=0}^H$ satisfying Table 1:*

*(i) For any node $s_h$ at depth $h \in \{0, \ldots, H\}$,*

$$\widehat{V}_n(s_h) \xrightarrow[n \to \infty]{\alpha_h,\beta_h} \widetilde{V}(s_h).$$

*(ii) For any node $s_h$ at depth $h \in \{0, \ldots, H-1\}$,*

$$\widehat{Q}_n(s_h, a) \xrightarrow[n \to \infty]{\alpha_{h+1},\beta_{h+1}} \widetilde{Q}(s_h, a), \quad \forall a \in \mathcal{A}_{s_h}.$$

*Proof.* We follow the proof technique of Dam et al. (2024b, Theorem 2).

**Base Case** ($H = 1$)**.** Consider the root node $s_0$. Each time we visit $(s_0, a)$, we collect:

- A reward sample $r^t(s_0, a)$ from the reward distribution $\nu_{s_0,a}^o$, which then leads to evaluating $\widehat{\nu}_n$ and $\widehat{R}_{s_0,a}^{\text{rob}} = \min_{r \in \widehat{\mathcal{R}}_{s_0,a}} \mathbb{E}_{R \sim r}[R]$, thus approximates the worst-case reward at $(s_0, a)$.

- A next state $s_1 \sim P^o_{s_0,a}$ from $M = |\mathcal{A}_{s_0}|$ possible states (denote such states as $S^1_0$). This then leads to $\widehat{p}_{s_0,a}$, and captures the worst-case value from the transition ambiguity set $\widehat{\mathcal{P}}_{s_0,a}$.

By definition of the robust Bellman backup, recall

$$\widetilde{Q}(s_0,a) = R^{\text{rob}}_{s_0,a} + \gamma\,\sigma_{\mathcal{P}_{s_0,a}}(\widetilde{V}),$$

where $R^{\text{rob}}_{s_0,a} = \min_{r_{s_0,a} \in \mathcal{R}_{s_0,a}} \mathbb{E}[r]$.

Since $H = 1$, the next state $s_1$ is treated as a leaf. We approximate $\widetilde{V}(s_1) \approx V_{\pi_0}(s_1)$, i.i.d. rollout returns under the policy $\pi_0$. By standard concentration bounds (e.g., Hoeffding), we obtain for all child nodes $s_1 \in S^1_0$:

$$\widehat{V}_n(s_1) \overset{\alpha_1,\beta_1}{\underset{n\to\infty}{\to}} \widetilde{V}(s_1). \tag{77}$$

Next, recall by equation 3:

$$\widehat{Q}_n(s_0,a) \leftarrow \widehat{R}^{\text{rob}}_{s_0,a} + \gamma\sigma_{\widehat{\mathcal{P}}_{s_0,a}}(\widehat{V}_{T_{s_1}(n)})$$

Here $\widehat{V}_{T_{s_1}(n)}$ is the estimated value at all child nodes $s_1 \in S^1_0$. By Lemma 17 and equation 77, it follows that

$$\widehat{Q}_n(s_0,a) \overset{\alpha_1,\beta_1}{\underset{n\to\infty}{\to}} \widetilde{Q}(s_0,a).$$

Since $s_0$ is the root node, we perform the power-mean backup on $\{\widehat{Q}_n(s_0,a)\}$:

$$\widehat{V}_n(s_0) = \Big( \sum_{a \in \mathcal{A}_{s_0}} \frac{T_{s_0,a}(n)}{n} \big[\widehat{Q}_{T_{s_0,a}(n)}(s_0,a)\big]^p \Big)^{\frac{1}{p}}.$$

Under Theorem 1 (from Dam et al. (2024b) for robust settings), we conclude

$$\widehat{V}_n(s_0) \overset{\alpha_0,\beta_0}{\underset{n\to\infty}{\to}} \widetilde{V}(s_0).$$

This establishes both points *(i)* and *(ii)* at depth 0 and confirms the result for $H = 1$.

**Inductive Step** $(H > 1)$. Assume the theorem holds for all search trees up to depth $H - 1$. We now add one more level to create a tree of depth $H$. Let $s_1$ be a child of the new root $s_0$. Then $s_1$ itself is a root of a subtree with depth $(H - 1)$. By the inductive hypothesis:

$$\widehat{V}_n(s_1) \overset{\alpha_1,\beta_1}{\underset{n\to\infty}{\to}} \widetilde{V}(s_1), \quad \widehat{Q}_n(s_1,a') \overset{\alpha_2,\beta_2}{\underset{n\to\infty}{\to}} \widetilde{Q}(s_1,a'), \forall a'.$$

At the new root $s_0$, we repeat the argument used in the base case:

- Observing rewards $r^t(s_0,a)$ from $\nu^o_{s_0,a}$.

- Transitioning under $P^o_{s_0,a}$ to state $s_1$.

Hence, Lemma 17 again implies

$$\widehat{Q}_n(s_0,a) \overset{\alpha_1,\beta_1}{\underset{n\to\infty}{\to}} \widetilde{Q}(s_0,a),$$

and the power-mean operator at $s_0$ yields

$$\widehat{V}_n(s_0) \overset{\alpha_0,\beta_0}{\underset{n\to\infty}{\to}} \widetilde{V}(s_0).$$

Thus, depth $H$ inherits the same concentration property from depth $(H - 1)$. This completes the inductive argument, establishing statements *(i)* and *(ii)* for any node at any depth $\leqslant H$.

$\square$

**Theorem 3.** *(Convergence of Expected Payoff)* *At the root node $s_0$, there is a choice of parameters yielding*

$$\left| \mathbb{E}\big[\widehat{V}_n(s_0)\big] - \widetilde{V}(s_0)\right| \leq \mathcal{O}\big(n^{-1/2}\big).$$

*Proof.* By Jensen's inequality (convexity of $|x|$), we obtain

$$\left| \mathbb{E}\big[\widehat{V}_n(s_0)\big] - \widetilde{V}(s_0)\right| \leq \mathbb{E}\left[\left|\widehat{V}_n(s_0) - \widetilde{V}(s_0)\right|\right]$$
$$= \int_0^\infty \mathbb{P}\Big(\big|\widehat{V}_n(s_0) - \widetilde{V}(s_0)\big| \geq s\Big)\,ds.$$

Next, we split this integral at $s = n^{-\alpha_0/\beta_0}$. Using the concentration property $\widehat{V}_n(s_0) \overset{\alpha_0,\beta_0}{\underset{n\to\infty}{\rightarrow}} \widetilde{V}(s_0)$, we have

$$\mathbb{P}\Big(\big|\widehat{V}_n(s_0) - \widetilde{V}(s_0)\big| \geq s\Big) \leq c_0\, n^{-\alpha_0}\, s^{-\beta_0},$$

for $s > n^{-\alpha_0/\beta_0}$. Hence,

$$\left| \mathbb{E}\big[\widehat{V}_n(s_0)\big] - \widetilde{V}(s_0)\right| \leq \int_0^{n^{-\frac{\alpha_0}{\beta_0}}} 1\,ds + \int_{n^{-\frac{\alpha_0}{\beta_0}}}^\infty c_0\, n^{-\alpha_0}\, s^{-\beta_0}\,ds$$
$$\leq n^{-\frac{\alpha_0}{\beta_0}} + c_0\, n^{-\alpha_0} \int_{n^{-\frac{\alpha_0}{\beta_0}}}^\infty s^{-\beta_0}\,ds$$
$$= n^{-\frac{\alpha_0}{\beta_0}} + \frac{c_0}{\beta_0 - 1}\, n^{-\alpha_0} \big[\, s^{-\beta_0+1}\big]_{s=n^{-\frac{\alpha_0}{\beta_0}}}^\infty.$$

Because $\frac{\alpha_0}{\beta_0} \leq \frac{1}{2}$ (see Theorem 1), the dominant term is $\mathcal{O}(n^{-\frac{1}{2}})$. Thus,

$$\left| \mathbb{E}\big[\widehat{V}_n(s_0)\big] - \widetilde{V}(s_0)\right| \leq \mathcal{O}\big(n^{-\frac{1}{2}}\big).$$

$\square$

## E. Experimental setup and Parameters selection

### E.1. Experimental setup

All experiments are done over 100 seeds, using $\gamma = 0.99$ and robustness budget $\rho = 0.5$, with these values showing consistent performance across preliminary experiments with different parameter settings. We use 2000 rollouts for The Gambler's Problem and 4000 rollouts for Frozen Lake.

We implement our robust MCTS framework by extending a base Monte Carlo Tree Search implementation from (Leurent, 2018). Our codebase adds Stochastic Power UCT and introduces new robust backup operators for handling different uncertainty sets (Total Variation, Chi-squared, and Wasserstein), while maintaining the core MCTS selection and expansion strategies. We also provide our code at `https://github.com/brahimdriss/RobustMCTS`.

### E.2. Environments

**The Gambler's Problem** (Sutton and Barto, 2018): a classic casino-inspired reinforcement learning environment where an agent starts with an initial capital and aims to reach a specific goal amount through a series of betting decisions. In our implementation, the agent begins with 50 units of capital and must reach a goal of 100 units to win. At each step, the agent can bet any amount up to its current capital. The environment has a win probability $p_h$ for each bet, where the agent either wins the wagered amount with probability $p_h$ or loses it with probability $1 - p_h$. The state space consists of all possible integer capital amounts from 0 to 100, with 0 and 100 being terminal states. The action space at each state includes all possible integer bets up to the current capital. This environment is particularly suitable for studying decision-making under uncertainty as it combines both risk management and optimal stopping aspects.

In our experiments, to reduce computational complexity while maintaining the same fundamental dynamics and challenges, we scaled down the problem to use a starting capital of 5 units and a goal of 10 units. This smaller scale version preserves all the essential characteristics and decision-making complexity of the original problem.

**Frozen Lake**(Towers et al., 2024): This environment presents a gridworld navigation challenge where an agent must traverse a 4x4 frozen surface from a starting position to a goal while avoiding holes. The surface is slippery, introducing stochastic dynamics where the agent's intended actions may result in sliding to adjacent states with some probability. The state space consists of 16 discrete states representing different positions on the grid, with some states marked as holes (H) and one goal state (G). The action space includes four possible movements: left, right, up, and down. When the agent executes an action, it moves in the intended direction with probability 1/3 and slides perpendicular to the intended direction (left or right) with probability 2/3, making the environment highly stochastic. This environment is particularly valuable for evaluating robust policies as it combines both navigational planning and uncertainty in action outcomes.

In our experiments, we define $p_{\text{slip}}$ as the probability that the executed action differs from the agent's selected action. When a slip occurs, the actual executed action is sampled uniformly at random, effectively modeling the uncertain dynamics of the frozen surface.

### E.3. Robust Performance Results

We investigate the impact of uncertainty budgets on agent performance in a modified gambler's problem. In this experiment, we fix the planning probability $p_h$ at $0.6$, the ambiguity set at Wasserstein. The agent's robustness is evaluated across different uncertainty budgets $\rho \in \{0.1, 0.3, 0.5, 0.7, 0.9\}$, where higher values of $\rho$ correspond to more conservative policies. For each uncertainty budget, we assess the agent's performance by varying the execution probability from 0.2 to 0.8, thus testing the policy's robustness to model misspecification. This experimental design allows us to analyze how different levels of conservatism (controlled by the uncertainty budget) affect the agent's ability to maintain performance when faced with discrepancies between planning and execution environments.

Figure 3 demonstrate a clear trade-off between performance and robustness across different uncertainty budgets. Agents with lower uncertainty budgets ($\rho = 0.1, 0.3$) achieve better performance when the execution probability matches or exceeds the planning probability, but their success rate drops significantly in misspecified environments. In contrast, higher uncertainty budgets ($\rho = 0.7, 0.9$) show more consistent performance across different execution probabilities, particularly maintaining better success rates when the execution probability is lower than the planning probability. This suggests that while conservative policies might not achieve optimal performance in well-specified environments, they provide better robustness to model misspecification. The moderate uncertainty budget ($\rho = 0.5$) appears to offer a balanced trade-off, maintaining reasonable performance across both regimes.

We now investigate a wide range of transition model ambiguities for the Frozen Lake environment. Table 4 provides an extended version of Table 2 with detailed success rates across different planning and execution probabilities. We observe that the performance of Stochastic-Power-UCT algorithm degrades faster for increased noise injection for slipping probabilities $p_{\text{slip}}$. We again see Wasserstein robust MCTS does well across all planning versus execution phases. All robust MCTS variants outperform the baseline.

Finally, our experiments reveal that the Wasserstein robust MCTS algorithm showcases the most robust performance across all variants. It might be of independent interest for future research to give a theoretical understanding of this phenomenon.

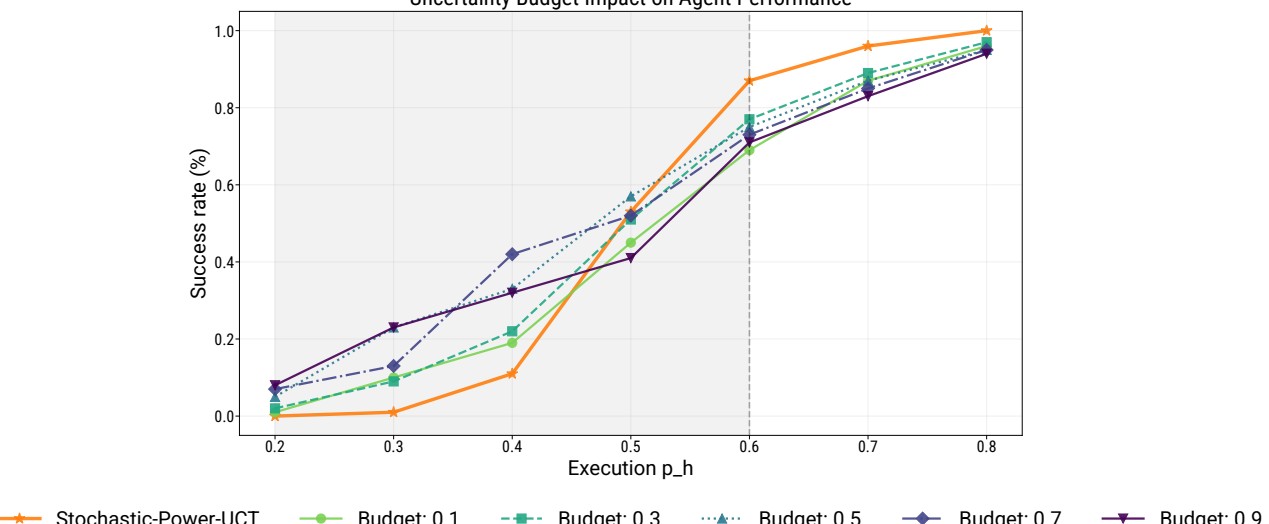

Figure 3: Performance comparison across different uncertainty budgets ($\rho$). Planning probability is fixed at $p_h = 0.6$ (vertical dashed line), while execution probability varies from 0.2 to 0.8. Higher uncertainty budgets lead to more conservative policies, showing improved robustness when $p_h \leqslant 0.6$ but potentially reduced performance when $p_h > 0.6$.

| Planning | | Execution $p_{\text{slip}}$ | | | | |
|---|---|---|---|---|---|---|
| $p_{\text{slip}}^{\text{plan}}$ | | 0.1 | 0.2 | 0.3 | 0.4 | 0.5 |
| 0.0 | Sp | **100** | 85 | 71 | **60** | 34 |
| | Tv | **100** | 84 | 71 | 51 | 39 |
| | Cs | **100** | **87** | 62 | 53 | 33 |
| | Ws | **100** | 86 | **72** | 58 | **45** |
| 0.1 | Sp | 65 | 52 | 41 | 32 | 21 |
| | Tv | 68 | 54 | 42 | 33 | 24 |
| | Cs | 95 | 82 | 65 | 52 | 35 |
| | Ws | **97** | **84** | **68** | **55** | **38** |
| 0.2 | Sp | 35 | 28 | 22 | 15 | 12 |
| | Tv | 38 | 30 | 25 | 18 | 15 |
| | Cs | 75 | 65 | **48** | 35 | 25 |
| | Ws | **78** | **68** | 45 | **38** | **28** |
| 0.3 | Sp | 15 | 12 | 10 | 8 | 7 |
| | Tv | 18 | 15 | 12 | 10 | 8 |
| | Cs | 55 | 45 | **35** | 25 | 18 |
| | Ws | **58** | **48** | 32 | **28** | **20** |
| 0.4 | Sp | 8 | 7 | 6 | 5 | 4 |
| | Tv | 10 | 8 | 7 | 6 | 5 |
| | Cs | 35 | 28 | 22 | 18 | 12 |
| | Ws | **38** | **30** | **25** | **20** | **15** |
| 0.5 | Sp | 5 | 4 | 4 | 3 | 3 |
| | Tv | 6 | 5 | 4 | 4 | 3 |
| | Cs | 25 | 20 | 15 | 12 | 8 |
| | Ws | **28** | **22** | **18** | **15** | **10** |

Table 4: Success rates (%) for planning with Power-UCT variants. Methods: Stochastic-Power-UCT (Sp), Robust version with Total Variation (Tv), Chi-squared (Cs), and Wasserstein (Ws) ambiguity sets. Underlined values indicate matching planning and execution $p_{\text{slip}}$. Bold indicates highest success rate per planning scenario.

