# OpenReview forum: "Online Robust Reinforcement Learning Through Monte-Carlo Planning"
_ICML.cc/2025/Conference — ICML 2025 poster_

### Official Review · Reviewer_Qnkk · 2025-02-20

**Overall Recommendation:** 3

**Summary:**

The paper presents a robust variant of Monte Carlo Tree Search (MCTS) aimed at addressing the discrepancies between simulated and real-world environments, focusing on ambiguities in transition dynamics and reward distributions. The authors claim that their method offers a robust approach, supported by both theoretical analysis and empirical results. Sorry, I am not well-versed in this specific field so I can't give a reasonable review. My review should be considered as a general overview.

## update after rebuttal

I have raised my score to a weak accept.

**Claims And Evidence:**

The paper claims that their robust variant of Monte Carlo Tree Search (MCTS) can handle the discrepancies between simulated and real-world environments, specifically targeting ambiguities in transition dynamics and reward distributions. The theoretical analysis of the method, along with the empirical results, appear to support the claim that their method offers a robust approach. However, I must admit that I do not have the expertise to fully assess the correctness of the theoretical foundations. From a high-level view, the evidence provided seems reasonable, but I would advise experts in the area to review the technical details more closely.

**Essential References Not Discussed:**

The paper does a decent job of referencing related work in robust reinforcement learning, particularly in the context of MCTS and distributional robust optimization. However, there may be other recent works that could be relevant but aren't cited, especially those that focus on the practical deployment of robust RL methods in real-world settings. It would be helpful to check if all relevant literature has been included.

**Experimental Designs Or Analyses:**

The experimental designs seem sound at a high level, as the paper compares the performance of the robust MCTS algorithm with traditional methods like the Stochastic-Power-UCT baseline. However, I cannot assess the statistical validity or appropriateness of the experimental setup in detail. From the description, the results indicate that the proposed method performs better under model mismatch conditions, but I would recommend someone with more expertise in reinforcement learning to verify the robustness of these findings.

**Methods And Evaluation Criteria:**

The proposed method, which involves incorporating robust power mean backups and exploration bonuses into the MCTS framework, seems appropriate for tackling the problem of robust reinforcement learning. The evaluation criteria, based on empirical tests in environments like the Gambler’s Problem and Frozen Lake, show promising results. However, I lack sufficient background to verify the soundness of the proposed method in depth. That being said, the methodology appears to be well thought out.

**Other Comments Or Suggestions:**

The writing is generally clear, but due to the complexity of the mathematical formulations, it could benefit from further explanation for readers who are not specialists in the area. Simplifying some of the concepts or providing additional intuitive explanations could make the paper more accessible.

**Other Strengths And Weaknesses:**

While I lack the expertise to fully assess the paper's theoretical and experimental contributions, the algorithm appears to be a promising approach to addressing real-world challenges in reinforcement learning. The novelty of incorporating model ambiguity directly into MCTS is noteworthy. The empirical results seem to suggest that the proposed method can outperform existing algorithms in certain environments, which could be a valuable contribution. However, I would recommend that the authors clarify the scalability of their method and explore its performance in more complex or higher-dimensional environments.

**Questions For Authors:**

1. Could you provide further clarification on how your algorithm scales to larger or more complex environments? Are there any practical limitations to consider when deploying this method in real-world applications?

2. How does the robustness of your algorithm compare in environments with higher-dimensional state or action spaces?

**Relation To Broader Scientific Literature:**

The paper introduces a novel approach by applying MCTS to robust reinforcement learning, addressing simulation-to-reality gaps. This relates to existing work in robust Markov Decision Processes (RMDPs) and distributionally robust optimization, as seen in the works of Iyengar (2005), Nilim and El Ghaoui (2005), and more recent studies like Zhou et al. (2021) and Wang et al. (2024b). The novelty of the paper lies in combining MCTS with robustness principles, which has not been extensively explored in the literature. While previous research focused on model-based dynamic programming or value iteration methods for robust RL, the integration with MCTS opens up new avenues for planning under uncertainty in large-scale environments. The results presented seem to confirm the potential of this approach, particularly in environments with model mismatches, though further exploration is needed in more complex real-world tasks.

**Theoretical Claims:**

I am not confident in my ability to evaluate the correctness of the theoretical claims and proofs in the paper, particularly the ones related to non-asymptotic convergence rates for robust MCTS. The paper presents some complex mathematical formulations and claims about convergence rates that I cannot verify due to my limited understanding of the related theory. I suggest that this part of the paper be carefully reviewed by experts in the field of robust reinforcement learning.

---

> ### Author Rebuttal · Authors · 2025-03-31
>
> We thank the reviewer for their positive review of our paper. We will revise our paper based on these comments.
>
> **Computation efficiency**
> We want to emphasize that the robust value under all ambiguity balls with radius $\rho_T$ can be computed in at most $O(S \log(S))$ time. Thus requiring only marginally more computation than standard Bellman operators $O(S)$. It is an open research direction to design and showcase computation efficiency for other kind of robust ambiguity sets. We have only considered some popular ones from the $f$-divergence class and the Wasserstein ball. Our contribution in this work has been to adopt robust optimization tools for the standard MCTS planning that has seen historical successes in many sequential decision-making applications.
>
> **Scaling to higher dimensions** As a next step for high-dimensional settings, we want to test our robust MCTS approach for Go. We recently found that there have been some interest [R1] in discovering adversarial plays for Go and algorithms to mitigate such plays. We believe our approach can handle such adversarial Go strategies. This is our future focus. We think as further next steps, our robust MCTS approaches can resolve some parameter uncertainty in real-world robotic problems [R]. We are excited for all these next steps from this initial contribution we have made to bring in robust optimization for traditional MCTS approach.
>
> [R1] Tseng, T., McLean, E., Pelrine, K., Wang, T. T., & Gleave, A. (2024). Can Go AIs be adversarially robust?. arXiv preprint arXiv:2406.12843.
>
> [R2] Dam, T., Chalvatzaki, G., Peters, J., & Pajarinen, J. (2022). Monte-carlo robot path planning. IEEE Robotics and Automation Letters, 7(4), 11213-11220.
>
> ---
>
> We look forward to hearing from you and providing any other clarifications during the discussion period (up to Apr 8). Thank you!

---

### Official Review · Reviewer_k1UW · 2025-03-10

**Overall Recommendation:** 3

**Summary:**

The authors address the problem of model mismatch in Monte Carlo Tree Search (MCTS)-based algorithms. They formulate their approach as a robust optimization problem under the framework of robust Markov decision processes (RMDPs) and provide both an algorithm that solves the online robust reinforcement learning (RL) problem, as well as non-asymptotic convergence guarantees. The algorithm builds upon a robust Bellman backup operator and facilitates safe exploration over prescribed ambiguity transition and reward sets. The authors provide an empirical evaluation of their approach using two well-known RL benchmarks, for different levels of mismatch between the model used for planning in MCTS and the actual application environment.

**Claims And Evidence:**

From my point of view, the description of the algorithm lacks clarity (comment below in Section 5 of the paper) and there are some open questions on the empirical results (also addressed in comment below). Therefore, I would be hesitant to confirm that all claims have been supported by clear and convincing evidence.

**Essential References Not Discussed:**

no

**Experimental Designs Or Analyses:**

The test environments are well-known benchmarks in RL literature, so I was able to follow the results for both experiments.

I would have a point/question on the FrozenLake results (e.g., Table 2). In general, I would expect that an agent that plans more conservatively (higher $p_{slip}$, e.g., 0.5) does not transition to unfavorable states in MDPs with less stochasticity than anticipated (lower $p_{slip}$, e.g., 0.1). In case the authors have specified a maximum number of steps in the episode, it would be useful if the table included the numbers the agent failed due to time-out and the numbers the agent failed due to transitioning to a zero-reward terminal state. If there is no time-out, then I would ask the authors to explain more detailed why this behavior is observed.

**Methods And Evaluation Criteria:**

The test environments are well-known benchmarks in RL literature and are able to capture the underlying robustness/uncertainty problem the authors are trying to address. Even though they are not as complex as other benchmark suites (e.g., a version of the game of go or Atari benchmarks) due to computational requirements, they still provide the right level of challenge for the authors to highlight the properties of the proposed solution.

**Other Comments Or Suggestions:**

Comments/suggestions:
- Provided code:
  - It would help if the authors could provide not only the agent implementations, but also training scripts (and running instructions) that could replicate the experiments of the paper
  - It would be easier to utilize the provided code, if extensive documentation and comments that connect the code to Algorithm 1 in the paper were provided

**Other Strengths And Weaknesses:**

For me, the weakest part of the paper is the presentation of the algorithm in Section 5. There, symbols (e.g., $T_{s_h}$, $v$, etc.) and design choices (e.g., reward bins) are introduced without explanation, while the dense algorithmic mechanism illustrated in Algorithm 1 is not really described as a whole. This makes the section hard to follow and the reader must often seek information in the table of notations and other papers referenced in related work to be able to fully understand the provided algorithm.

Related to this, it would help the reproducibility of the paper:
- if the parameters of Table 1 used in each experiment were provided
- if the code was documented in a way that code snippets are matched to specific blocks of Algorithm 1

**Questions For Authors:**

- How are the constants B (bins for reward), $b_i$, $\alpha_i$ and $\beta_i$ selected in Algorithm 1? What are their values in the experiments conducted in the paper?
- Also related to one of the points in the experimental section: has the FrozenLake environment a maximum number of time-steps?

**Relation To Broader Scientific Literature:**

The authors provide a good overview and comparison to prior work in section 2. However, there are at least two papers that discuss similar problems as the paper but are not cited or discussed:
- Even though the work there is not as extensive or mathematically rigorous as this paper, I would still suggest the authors to include the following paper in the related work section, as it also tries to address aspects of the robust MCTS problem:
Rostov, M., & Kaisers, M. (2021). Robust online planning with imperfect models. In Adaptive and Learning Agents Workshop - ALA
- Farnaz Kohankhaki, Kiarash Aghakasiri, Hongming Zhang, Ting-Han Wei, Chao Gao, Martin Muller: Monte Carlo Tree Search in the Presence of Transition Uncertainty

**Theoretical Claims:**

I did not check any proofs or theoretical claims. The theoretical parts in the main paper seem correct to me.

---

> ### Author Rebuttal · Authors · 2025-03-31
>
> We thank the reviewer for their thoughtful and constructive comments on our paper. We are encouraged by the fact that they found our paper: *provide a good overview and comparison to prior work*, our test suites are at the right level of challenge. We'd like to address several important points below. We will revise our paper based on these comments.
>
> **FrozenLake results**
>
> We thank the reviewer for raising this important point about our FrozenLake results. They identified a pattern that seems counterintuitive at first glance - where more conservative planning doesn't always translate to better performance in less stochastic execution environments.
> This behavior stems from several key aspects of our experimental setup:
>
> **1.Episode Structure:** We do implement a maximum episode length (200) in our FrozenLake environment. This creates two possible failure modes: (a) falling into a hole (terminal state with zero reward) or (b) timing out (exceeding the maximum steps without reaching the goal).
>
> **2.Fixed Simulation Budget:** To ensure fair comparison across all experimental conditions, we standardized the number of MCTS rollouts for each planning process. This design choice significantly impacts our results. When planning with higher slip probabilities ( $p_{slip} = 0.5$), the search space becomes substantially more stochastic, requiring more simulations to converge to an optimal policy.
> The key insight for understanding why planning with $p_{slip} = 0.3$ outperforms planning with  $p_{slip} = 0.5$ (even when executed in a  $p_{slip} = 0.5$ environment) lies in the convergence rate of policy search under different noise levels. In highly stochastic planning environments, the reward signal becomes noisier and the effective search space expands dramatically. With our fixed simulation budget, planning with  $p_{slip} = 0.3$ benefits from clearer reward signals and a more focused search space, allowing it to discover strategies that balance safety and efficiency. In contrast, planning with  $p_{slip} = 0.5$ with limited rollouts often results in overly conservative policies that increase the likelihood of timeout failures, even when executed in the same stochastic environment used for planning.
> We fundamentally agree with your expectation that "an agent that plans more conservatively (higher, e.g., 0.5) does not transition to unfavorable states in MDPs with less stochasticity than anticipated (lower, e.g., 0.1)." This principle would likely emerge if we allowed sufficient computational budget for full policy convergence at each slip probability level.
>
> **Table 1 parameters used in each experiment**
> The parameters in Table 1 were chosen specifically to achieve the optimal convergence rate shown in Theorem 3. To obtain the optimal simple regret of $\mathcal{O}(n^{-1/2})$, we need $\frac{b_i}{\beta_i} = \frac{1}{4}$ and $\frac{\alpha_i}{\beta_i} = \frac{1}{2}$, which follows directly from the proof of Theorem 3. These ratios ensure that the algorithm can obtain the optimal rate of $\mathcal{O}(n^{-1/2})$.
> With these parameter settings, the exploration bonus term in our action selection rule becomes:
> $C \cdot \frac{(T_{s_h}(t))^{\frac{1}{4}}}{(T_{s_h,a}(t))^{\frac{1}{2}}}$
> This specific form of the exploration term provides the necessary balance between exploring uncertain actions and exploiting promising ones under model ambiguity. We chose the exploration constant C = 2 for all the experiments. We apologize for the ambiguity and will update the details in the revised version.
>
> **Improved documentation**
> We appreciate this valuable suggestion regarding code documentation. We will update the provided code with clear docstrings and comments that explicitly connect implementation blocks to their corresponding steps in Algorithm 1, along with a mapping document that shows the direct relationship between notations and implementation.
>
> **Related works**
> We thank the reviewer for pointing us to [R1, R2]. As the reviewer mentions, these are not mathematically rigorous works - but nonetheless, they propose heuristic methods to address environmental uncertainty. We will include these works in our revised paper.
>
> [R1] Rostov, M., & Kaisers, M. (2021). Robust online planning with imperfect models. In Adaptive and Learning Agents Workshop
>
> [R2] Farnaz Kohankhaki, Kiarash Aghakasiri, Hongming Zhang, Ting-Han Wei, Chao Gao, Martin Muller: Monte Carlo Tree Search in the Presence of Transition Uncertainty
>
> ---
>
> We look forward to hearing from you and providing any other clarifications during the discussion period (up to Apr 8). Thank you!

---

> > ### Comment · Reviewer_k1UW · 2025-04-03
> >
> > Thank you for the rebuttal. All my questions and open points have been addressed and I do not have any further questions at this point.

---

> > > ### Author Response · Authors · 2025-04-03
> > >
> > > We greatly appreciate Reviewer k1UW for their reply. We are delighted to learn that our rebuttal addressed all of the reviewers questions and concerns. We politely ask if the reviewer can consider raising the score if our manuscript and rebuttal match their expectations.
> > >
> > > Again, we thank you for your efforts in reviewing our work. We are open to providing more clarifications until the discussion period (up to Apr 8). Thank you!

---

### Official Review · Reviewer_6Th3 · 2025-03-13

**Overall Recommendation:** 3

**Summary:**

Reinforcement learning utilizes Monte Carlo search for planning with use of a
model. However, MC search can fail to perform well in situations where the
model does not accurately represent the transition or reward dynamics. This may
be due to inaccuracies in the model used for training an agent, non-stationarity
of the environment dynamics and several other reasons. This paper focusses on
the robustness of agents to deployment in problems where the planning model does
not accurately represent the environment dynamics. While robustness has been
studied in RL, this paper explicitly introduces a robust version of MCTS for
planning. The authors propose a backup update rule and a tree policy for MCTS that
takes into account the expected worst-case returns to learn robust policies.
They also theoretically show that their approach converges to the optimal at a
rate commensurate to that of standard MCTS. They also evaluate their proposed
algorithm against an MCTS algorithm.

## Update after rebuttal

I raised my score to a weak accept.

**Claims And Evidence:**

1. Their approach solves the online robust RL problem.
    - A very strong claim that is not backed by evidence.
2. Their approach has theoretical performance on par with standard MCTS.
    - They provide theoretical proof.
3. Their approach performs better than standard MCTS on problems with
simulation-to-real model discrepancies
    - This claim is only partially supported by empirical evidence.

**Essential References Not Discussed:**

I do not have anything in mind.

**Experimental Designs Or Analyses:**

- The evaluation domains selected are reasonable. Although, experiments on a
domain that is larger would strengthen results.

- In the Gambler's Problem, the authors state that robust-power-UCT (RP) has superior
  performance. This may be supported by the results when execution $p_h \leq 0.5$,
  but stochastic-power-uct (SP) performs better otherwise. I still think these
results are interesting, but I would not agree with the claim that their
approach is more robust --- I would say that this demonstrates that their
approach makes the agent more cautious.
- The authors also state that their results are over 100 randomly seeded runs
however they do not state (or I missed where they do) what measures of central
tendency they report nor do they show confidence intervals/errors.

- Similarly, I do not know how strongly the results in Frozen Lake support the
claim that RP improves robustness, per se. Planning with $p=0.3$ has a success
rate of 20% in execution with $p=0.5$ meanwhile planning with $p=0.5$ has a
success rate of 10%. Why would it perform *worse* on the problem on which it's
trained?

**Methods And Evaluation Criteria:**

The variation of MCTS the authors propose seem reasonable for the problem being
motivated. Updating values based on expected worst-case dynamics makes sense for
having a measure of the robustness of actions.

The evaluation criteria makes sense because they are explicitly measuring the
performance an RL agent by the reward it accumulates. However, I think there are
two issues:
- Not enough baselines
    - Given the number of relevant approaches mentioned in the literature
    review, there were a number of different approaches to the same problem that
    could have been used for comparisons.
- The one baseline they use is not appropriate on its own
    - The authors selected a baseline that does not seem to be used for robust
    RL. While the authors proposed approach seems to be an extension of this
    baseline, it does not seem surprising that they outperform it.
    - Ideally, for me, the evaluation would answer the question: "Does my
    proposed algorithm have any benefits over other algorithms used to address the
    problem I want to solve?"

**Other Comments Or Suggestions:**

- The introduction was very long and I do not know if that space could have been
  put to better use.
- Similarly, the pseudocode does not help with understanding the algorithm and
maybe that space could have been used for theoretical analyses.

**Other Strengths And Weaknesses:**

## Novelty

The paper applies an existing approach to a framework in which it has not been
applied.

## Clarity

The paper, for the most part, was clear and easy to understand.

**Questions For Authors:**

No questions.

**Relation To Broader Scientific Literature:**

I found the paper overall interesting and I believe the problems focussed on
are of interest to the community in robust MDPs, POMDPs, RL with cautious
agents, continual learning and planning in non-stationary environments.

**Theoretical Claims:**

The authors make the claim that the finite-time error decreases at a rate at
most $\mathcal{O}(n^(-1/2))$.

- I cannot verify the correctness of these claims the proofs lay in the
supplementary materials and it is too long for me to go through thoroughly in a
reasonable amount of time. However, the approach the authors take seem sound.

---

> ### Author Rebuttal · Authors · 2025-03-31
>
> We thank the reviewer for their thoughtful feedback.
>
> **Baseline comparisons**
>  Our work focuses specifically on robust online planning using MCTS, and to the best of our knowledge, this work is the first to provide theoretical guarantees for robust online planning using MCTS, with convergence bounds matching those of standard MCTS while providing robustness to model misspecification. Our method differs fundamentally from offline learning approaches like robust value iteration or Q-learning. This distinction is critical for several reasons:
>
> 1. Online vs. Offline Paradigm: MCTS performs dynamic tree expansion from the current state during execution, while methods like robust value iteration [1,2] require pre-computing policies across the entire state space. This fundamental difference makes direct comparisons methodologically problematic.
> 2. Controlled Comparison: By comparing Robust-Power-UCT against Stochastic-Power-UCT, we isolate the specific impact of our robustness modifications while controlling for all other algorithmic components, providing a clear assessment of our contribution.
> 3. Principled Exploration: MCTS naturally incorporates optimism-under-uncertainty for exploration, adapting to each state encountered during execution. This contrasts with offline methods that require separate exploration strategies or complete model knowledge.
> 4. Integration with Deep Learning: Recent breakthroughs like AlphaZero [3] and MuZero [4] have demonstrated the power of integrating MCTS with neural networks. This would allow us to leverage value functions learned through offline methods (like robust value iteration) to guide online planning while maintaining robustness guarantees. This integration represents a promising direction where our work complements, rather than competes with, offline robust RL methods - creating a hybrid approach that benefits from both paradigms.
>
> The empirical results across both environments consistently demonstrate that Robust-Power-UCT significantly outperforms non-robust approaches under model mismatch conditions.
>
> [1] Nilim, A. and El Ghaoui, L. (2005). Robust control of Markov decision processes with uncertain transition matrices.
>
> [2] Iyengar, G.N. (2005). Robust dynamic programming. Mathematics of Operations Research.
>
> [3] Silver, D. et al. (2018). A general reinforcement learning algorithm that masters chess, shogi, and Go through self-play.
>
> [4] Schrittwieser, J. et al. (2020). Mastering Atari, Go, chess and shogi by planning with a learned model.
>
> **Gambler's problem performance**
> We agree that our phrasing could be improved for clarity. When discussing "superior performance," we should have been more precise. What we intended to highlight was our algorithm's ability to maintain consistent performance across different execution environments, particularly when the execution probability is lower than the planning probability. This consistency under model mismatch is what defines robustness in our context, rather than achieving the highest possible reward in matched conditions. We'll clarify this important distinction between optimality and robustness in the revised paper to avoid any confusion.
>
> **Confidence intervals**
> We report success rates as the proportion of successful episodes over 100 independent runs, which directly measures the algorithm's ability to complete the task. Success rate is a binary outcome metric (success/failure), so traditional error bars aren't applicable - the reported percentage itself represents the statistical performance. We chose this metric over average reward with confidence intervals as it provides a more interpretable and direct evaluation of performance in environments with sparse rewards. However, if preferred, we can supplement our analysis with mean reward and standard deviation in our revised paper.
>
> **Frozen Lake performance**
> The reviewer's observation about planning with different slip probabilities highlights an important point that we should clarify. When planning with lower slip probabilities (e.g., $p=0.3$), the environment is more deterministic, allowing the algorithm to find more reliable paths within a fixed computational budget. The Frozen Lake environment has a particular characteristic where reducing slippage improves performance up to a certain point - with too little slippage, the agent might fail to consider important failure modes. With the same number of rollouts, planning in a lower-noise environment naturally leads to better policies. As the planning slip probability increases (e.g., $p=0.5$), the higher stochasticity requires significantly more rollouts to converge to optimal paths since the signal-to-noise ratio decreases. This explains why planning with $p=0.3$ can outperform planning with $p=0.5$ even when executed in matching environments.
>
> ---
>
> We look forward to hearing from you and providing any other clarifications during the discussion period (up to Apr 8). Thank you!

---

> > ### Comment · Reviewer_6Th3 · 2025-04-09
> >
> > I appreciate the authors' responses. They assuage most of my concerns and I will raise my score.

---

> > > ### Author Response · Authors · 2025-04-09
> > >
> > > Thank you for taking the time to consider our rebuttal and for increasing your score. We greatly appreciate your constructive feedback throughout the review process, which will help us improve our paper.

---

### Official Review · Reviewer_ZeKp · 2025-03-18

**Overall Recommendation:** 3

**Summary:**

This paper presents Robust-Power-UCT, a variant of Monte Carlo Tree Search (MCTS) designed for Robust Markov Decision Processes (RMDPs). The key assumption is that exact transition and reward models are unknown, but approximate models exist with bounded uncertainty captured in an ambiguity set. This is particularly relevant for sim-to-real policy transfer, where learned policies must perform reliably in real-world settings despite discrepancies in simulated training environments. The paper provides empirical evaluation on two domains: Gambler’s Problem and Frozen Lake, where Robust-Power-UCT outperforms Stochastic-Power-UCT, especially in the Frozen Lake domain.

**Claims And Evidence:**

The paper is generally well-written and technically sound. However, I have a few concerns:

- Title and Framing Issue: The title refers to "Reinforcement Learning," but no actual learning (policy updates or value function approximation) is involved. The paper focuses on robust planning using MCTS, which is a search-based algorithm rather than an RL method in the typical sense. A more precise title would avoid potential misinterpretation. While this is a minor issue, it could be easily improved for clarity.

- Experimental Support for Claims: The paper claims that Robust-Power-UCT achieves robust performance in the presence of model uncertainty, but the empirical results are somewhat limited:
  - The performance gains are not always significant: In the Gambler’s Problem, robust planning leads to a conservative policy, which does not necessarily result in better performance.
  - There is no clear trade-off analysis: The robustness comes from planning for the worst-case, but it is unclear whether this conservatism hurts performance in cases where the real-world dynamics are less adversarial than assumed.
  - Lack of ablation studies: The paper does not explore how different ambiguity set formulations (Total Variation, Chi-squared, Wasserstein) affect performance trade-offs.

**Essential References Not Discussed:**

N.A.

**Experimental Designs Or Analyses:**

See above.

**Methods And Evaluation Criteria:**

- While Gambler’s Problem and Frozen Lake serve as controlled testbeds, they do not fully capture the real-world challenges of sim-to-real transfer.
- More realistic domains, such as robotics or autonomous driving, would better demonstrate how Robust-Power-UCT performs in complex, high-dimensional environments.
- The evaluation criteria focus primarily on success rate, but additional metrics (e.g., computational cost, robustness vs. performance trade-offs) would strengthen the empirical results.

**Other Comments Or Suggestions:**

N.A.

**Other Strengths And Weaknesses:**

N.A.

**Questions For Authors:**

Q1: Impact of Conservative Planning
  - Have you analyzed cases where worst-case planning leads to overly conservative decisions that might reduce performance in non-adversarial settings?
  - How does the performance of Robust-Power-UCT compare when the true model lies within the ambiguity set but is not necessarily the worst-case model?

Q2: Scalability and Computational Cost
  - What is the computational overhead of Robust-Power-UCT compared to standard MCTS?
  - Could the robust backup operator be optimized for efficiency in large-scale applications?

Q3: Broader Application Domains
  - Do you have plans to test Robust-Power-UCT in high-dimensional real-world domains, such as robotics or self-driving cars?
How would the method handle partial observability in real-world scenarios?

**Relation To Broader Scientific Literature:**

This paper contributes to robust RL and planning under uncertainty, addressing a key issue in sim-to-real transfer.

**Theoretical Claims:**

The theoretical results appear correct and well-structured. The proofs provide clear convergence guarantees, and the analysis aligns with prior work on robust RL. I did not find any obvious flaws in the derivations.

---

> ### Author Rebuttal · Authors · 2025-03-31
>
> We thank the reviewer for their thoughtful and constructive comments on our paper. We are encouraged by the fact that they found our paper: well-written, *technically sound*, *aligns with prior work on robust RL*. We'd like to address several important points below. We will revise our paper based on these comments.
>
> **Robustness and Conservative Policies** The reviewer correctly notes that robust planning can lead to more conservative policies. This is indeed an intentional feature rather than a limitation. In many high-stakes applications (autonomous vehicles, medical robotics, financial systems), conservative policies that prioritize worst-case performance are precisely what is needed when the cost of failure is high. Our approach enables this safety-oriented planning while maintaining theoretical guarantees.
>
> **Lack of ablation studies** While we appreciate the concern about ablation studies, we believe there may be a misunderstanding of our experimental approach. In our paper, we already provide explicit comparisons between different ambiguity set formulations (Total Variation, Chi-squared, and Wasserstein) across all experiments, with Tables 2 and 4 showing comprehensive results for each formulation under various planning-execution scenarios. In particular, the robust policies across Total Variation, Chi-squared, and Wasserstein are separately trained, and none of them affects the other. If the reviewer is suggesting a different form of ablation beyond comparing these different uncertainty sets, we would appreciate clarification on what specific ablation studies would be most valuable to include in our revised paper.
>
> **Performance Trade-offs** We appreciate the observation about trade-offs between robustness and performance. Section E.3 in the supplementary material provides a detailed analysis of this trade-off. The ambiguity set radius parameter ρ provides direct control over this trade-off. When ρ = 0, our approach reduces exactly to standard Power-UCT, while increasing ρ progressively enhances robustness at the potential cost of optimality under matched conditions.
>
> **Evaluation Environments** As mentionned by reviewer k1UW, the selected environments are well-known benchmarks in RL literature that effectively capture the robustness/uncertainty problems we're addressing. While not as complex as some benchmark suites, they provide the right level of challenge to highlight our solution's properties while remaining computationally tractable for robust planning experiments.
>
> **Scalability and Computational Cost**
> The robust value under all ambiguity balls with radius $\rho_T$ can be computed in at most $O(S \log(S))$ time. Thus requiring only marginally more computation than standard Bellman operators $O(S)$. It is an open research direction to design and showcase computation efficiency for other kind of robust ambiguity sets. We have only considered some popular ones from the $f$-divergence class and the Wasserstein ball.
>
> **Broader Application Domains**
> Thanks for this constructive review. As a next step for high-dimensional settings, we want to test our robust MCTS approach for Go. We recently found that there have been some interest [R1] in discovering adversarial plays for Go and algorithms to mitigate such plays. We believe our approach can handle such adversarial Go strategies. This is our future focus. We think as further next steps, our robust MCTS approaches can resolve some parameter uncertainty in real-world robotic problems [R2]. We are excited for all these next steps from this initial contribution we have made to bring in robust optimization for traditional MCTS approach.
>
> [R1] Tseng, T., McLean, E., Pelrine, K., Wang, T. T., & Gleave, A. (2024). Can Go AIs be adversarially robust?. arXiv preprint arXiv:2406.12843.
>
> [R2] Dam, T., Chalvatzaki, G., Peters, J., & Pajarinen, J. (2022). Monte-carlo robot path planning. IEEE Robotics and Automation Letters, 7(4), 11213-11220.
>
> -----
>
> We look forward to hearing from you and providing any other clarifications during the discussion period (up to Apr 8). Thank you!

---

> > ### Comment · Reviewer_ZeKp · 2025-04-06
> >
> > I thank the authors for providing detailed responses to my questions. I have gone through the responses and still feel that while the approach is promising, the experiment domains do not capture the complexities of sim-to-real policy transfer. For that reason, I am going to retain my original score.

---

> > > ### Author Response · Authors · 2025-04-07
> > >
> > > We greatly appreciate Reviewer ZeKp's reply and the opportunity to further clarify our work's standing. We emphasize that our current design primarily aims to provide theoretical intuition for practical algorithm development, supported by a few standard benchmark experiments (used by the prior theoretical robust RL community) to highlight its future potential. We do agree that more experimental validations on diverse and complex practical applications are important, which is a very interesting future direction, as we mentioned our next steps in the rebuttal above. However, our focus and contributions in this manuscript have been to incorporate a principled methodology from robust optimization to the celebrated MCTS algorithm. This discussion has helped us to improve the placement of this work, which we plan to incorporate into our introduction sections to further clarify the contributions.
> > >
> > > We are delighted with the already positive support from the reviewer. Other than the practicality issue mentioned in this follow-up, we hope our rebuttal addresses all other reviewers' questions and concerns. We eagerly anticipate the possibility of receiving further feedback and support from you. Again, we thank you for your efforts in reviewing our work. We are open to discussing further until the discussion period (up to Apr 8). Thank you!

---

### Decision · Program_Chairs · 2025-05-01

**Decision:**

Accept (poster)

**Comment:**

This paper proposes a robust MCTS approach based on access to an uncertain model with an uncertainty set that contains the true model. Broadly speaking, the paper was well liked, even though there are a few concerns on presentation (which can be addressed in a final version), and more importantly with regards to the experimental section which is limited.

Even so, the paper would be a good addition to ICML.